# The Use of Anodic Oxides in Practical and Sustainable Devices for Energy Conversion and Storage

**DOI:** 10.3390/ma14020383

**Published:** 2021-01-14

**Authors:** Janaina Soares Santos, Patrícia dos Santos Araújo, Yasmin Bastos Pissolitto, Paula Prenholatto Lopes, Anna Paulla Simon, Mariana de Souza Sikora, Francisco Trivinho-Strixino

**Affiliations:** 1Department of Physics, Chemistry and Mathematics, Federal University of São Carlos (UFSCar), Via João Leme dos Santos Km 110, Sorocaba 18052-780, Brazil; janasoares@ymail.com (J.S.S.); patricia.araujo@estudante.ufscar.br (P.d.S.A.); yasminbp@estudante.ufscar.br (Y.B.P.); paula.prenholatto@estudante.ufscar.br (P.P.L.); 2Department of Chemistry, Universidade Tecnológica Federal do Paraná (UTFPR), Via do Conhecimento Km 1, Pato Branco 85503-390, Brazil; asimon@alunos.utfpr.edu.br (A.P.S.); marianasikora@utfpr.edu.br (M.d.S.S.); 3Chemistry Graduate Program, Campus CEDETEG, Midwestern Parana State University (UNICENTRO), Alameda Élio Antonio Dalla Vecchia, Guarapuava 85040-167, Brazil

**Keywords:** anodic oxides, nanostructures, anodization, dye-sensitized solar cells, PEC water-splitting, fuel cell, supercapacitors, batteries

## Abstract

This review addresses the main contributions of anodic oxide films synthesized and designed to overcome the current limitations of practical applications in energy conversion and storage devices. We present some strategies adopted to improve the efficiency, stability, and overall performance of these sustainable technologies operating via photo, photoelectrochemical, and electrochemical processes. The facile and scalable synthesis with strict control of the properties combined with the low-cost, high surface area, chemical stability, and unidirectional orientation of these nanostructures make the anodized oxides attractive for these applications. Assuming different functionalities, TiO_2_-NT is the widely explored anodic oxide in dye-sensitized solar cells, PEC water-splitting systems, fuel cells, supercapacitors, and batteries. However, other nanostructured anodic films based on WO_3_, Cu_x_O, ZnO, NiO, SnO, Fe_2_O_3_, ZrO_2_, Nb_2_O_5_, and Ta_2_O_5_ are also explored and act as the respective active layers in several devices. The use of AAO as a structural material to guide the synthesis is also reported. Although in the development stage, the proof-of-concept of these devices demonstrates the feasibility of using the anodic oxide as a component and opens up new perspectives for the industrial and commercial utilization of these technologies.

## 1. Introduction

This review compiles the recent advances (2015–2020) using anodic oxides as part of energy conversion and storage devices. It presents studies that bring applied technological aspects that are being developed or are already in the implementation stage and does not present an in-depth discussion regarding anodic oxide materials’ synthetic methodology. The fabrication of anodic oxides from anodization of metallic substrates offers several advantages that can impact the technology economically in biological, environmental, and energy fields. The anodic oxidation is a facile, low-cost, and scalable method to obtain nanostructured porous oxide films with large surface areas by strict control of the morphological and structural properties, such as tube length, pore size, film thickness, and rigid adhesion to the substrate [1,2,3]. These features make the anodic films, such as the popular TiO_2_ nanotubes (TiO_2_-NT) and anodic aluminum oxide (AAO), attractive for the development of a wide range of technologies, including drug delivery systems [4], implants [5], sensors [6], filtration membranes [7], photocatalysts for pollutant degradation [8], energy conversion [9,10,11] and energy-storage systems [12,13], as illustrated in Scheme 1.

In the context of global sustainable strategies used to minimize the problems caused by industrialization and population growth, scientists have made significant efforts to develop advanced materials and systems based on clean processes that exploit renewable energy sources. In this sense, solar energy, a “green” resource, is essential as a substitute for conventional and non-renewable energy sources due to its abundance, availability, and direct harvest. The solar light can be directly converted into electric energy by photovoltaic (PV) devices such as solar cells. On the other hand, the photoelectrochemical (PEC) devices operate using solar energy or another light source to produce hydrogen gas from water molecules. Considered as a green fuel, hydrogen can be stored or converted into electric energy in fuel cells.

Therefore, PV, PEC, and hybrid PV–PEC systems are considered attractive technologies for sustainable production of electric energy and renewable fuel. One of the main challenges in this sector is to improve the efficiency and performance of these devices by the fabrication of stable photoelectrodes with high photoconversion rates. In general, TiO_2_-NT is the preferred anodic oxide explored in dye-sensitized solar cells and PEC water-splitting systems [10,17,18,19]. Due to its photocatalytic properties, chemical stability, and unidirectional orientation, this semiconductor is frequently used as the photoanode; exceptions include reports of WO_3_ [20,21] and Fe_2_O_3_ [22]. The use of copper oxides as photocathodes is emerging [23,24]. The utilization of AAO as a template or scaffold is also addressed [2,25,26].

Considering devices that operate essentially via electrochemical processes, the most promising sustainable systems are the fuel cell (for energy conversion) and the supercapacitors and rechargeable batteries (for energy storage). In these applications, TiO_2_-NT is also the widely explored anodic oxide, but in this case, the properties of interest are its chemical stability, low-cost, and high surface area, despite its low conductivity [27,28]. In fuel cells, this oxide is employed as support for the catalyst in both anodes [29] and cathodes [30] to increase the electrodes’ stability. In supercapacitors, the excellent adhesion between anodic oxide and metal is advantageous for the fabrication of electrodes without a binder agent, reducing the synthesis steps [31]. In rechargeable batteries, large surface areas with open frameworks for insertion of ions are the aim. Additionally, the low volume expansion during the Li^+^ insertion/extraction is an advantage that can increase the stability and safety of the Li-ion battery [32]. Besides TiO_2_, the interest in the properties of other nanostructured anodic films for energy storage systems, such as Cu(OH)_2_, NiO, WO_3_, SnO, and Ta_2_O_5_, has been increasing in the field.

Herein, we provide an overview and state-of-the-art use of anodic oxides in energy conversion and storage devices that encompass the scientific and technological aspects of the Sustainable Development Goals established by the United Nations [33]. This review focuses on applying the anodic oxides and their main contributions in improving the efficiency and performance of the devices for energy conversion and storage applications. For this, we selected papers that, in addition to investigating the materials’ properties, also explored the materials in practical applications. Papers essentially describing synthetic routes, a material’s characterization, or the investigation of a material’s properties are not addressed in this review. For such topics, we recommended the excellent review papers in the anodic oxide field [2,8,34,35,36,37,38,39,40].

Initially, we present the common aspects of the electrochemical synthesis of these materials, i.e., the experimental conditions frequently employed for the anodization and heat treatment. According to each application, the strategies adopted to modify and tune the material’s properties will be presented in the following sections. In sequence, we will describe the uses of anodic oxides in devices for energy conversion that operate via photo and photoelectrochemical processes: the PV and PEC systems. We will then present the studies describing devices used for energy conversion and storage produced via electrochemical processes: the fuel cells, supercapacitors, and batteries.

## 2. General Aspects of Anodic Oxide Synthesis for Energy Applications

Nanostructured anodic oxides are an excellent alternative for developing energy devices’ components due to a series of specific advantages present in each energy application. However, we can summarize the general advantages of their utilization as follows:The anodic oxides are generally synthesized in environmentally friendly experimental conditions (mild temperature synthesis and low toxicity substances applied);They offer facile control of synthesis parameters, such as domain morphology, composition, and structure with the potential to anchor specific catalyst substances to use them as anode or cathode based-materials;The materials present a high surface area per volume;They allow facile modulation of nanostructure architecture to enhance ion transport;The mostly anodic oxides are chemically stable;In most cases, they offer excellent adhesion between different layers, avoiding binder agents.

The synthetic routes employed to fabricate the anodic oxides generally involve the following steps: (i) pre-treatment of the substrate, (ii) anodization, (iii) heat treatment, and (iv) surface modification if necessary. This section depicts the common aspects of the syntheses of TiO_2_, Al_2_O_3_, WO_3_, Fe_2_O_3_, ZnO, Cu_x_O, ZrO_2_, NiO, SnO, Nb_2_O_5_, and Ta_2_O_5_, which include the anodization procedure to grow the anodic oxides, followed by heat treatment procedure for oxide modification or crystallization.

The pre-treatment steps (cleaning, polishing, pre-annealing procedures) will not be described here because they can vary according to the metal and purpose. For this, we recommend consulting the original papers for details. Furthermore, since some functionalities were designed to attend to the specific properties of the device´s materials, the additional steps employed for surface modification of the oxide will be described in the following sections with each material’s application description.

The anodization consists of an oxidation process of a metallic substrate (M^o^ → M^z+^ + ze^−^) working as the anode of an electrochemical reactor in a proper electrolyte. The resulting anodic oxide film grown over the metal substrate can form a rigid compact layer or a nanostructured layer. The former case refers to barrier-type oxide films [2]. The latter case involves the formation of nanostructured oxide films with different morphologies, such as nanopores, nanotubes, nanowires, nanorods, nanopetals, and other nanogeometries [1,2,3,41,42]. The type of oxide formed depends on the experimental conditions. The main parameters that must be considered to control the morphology and composition of the anodic film during the anodization step are:Substrate: composition, purity, rugosity, and surface defects.Electrolyte: composition, temperature, and stirring.Electrical parameters: galvanostatic, potentiostatic, potentiodynamic, pulsed, or hybrid methods.Synthetic route: one-step, two-step or multi-step anodization.Anodizing time.

The mechanisms of growth of anodic oxide nanostructures and the influences of anodizing conditions on the oxide film’s final properties are well-known in the scientific literature, with many papers devoted to these topics having been published in the last decades [1,2,3,43,44,45]. In general, to form a nanostructured morphology with a large surface area, the anodization must be performed in an electrolyte in which the formed oxide is partially soluble, for instance, oxalate media to produce nanoporous alumina [2,45] or fluoride ions to grown nanotubes of TiO_2_ or ZrO_2_ [1,46,47]. The pore or nanotube diameter can be controlled by the applied potential and temperature, whereas the oxide layer’s thickness can be tuned by the anodizing time [48,49]. The anodizing time can vary from a few minutes to several hours depending on the layer’s desired thickness, usually obtained in the micrometer range [2,49,50]. Polishing procedures and the use of high-purity substrates tend to increase the orderly arrangement of the nanostructure. The two-step anodization method can be applied to the production of highly ordered nanotube/nanopore arrays [2,3]. This procedure uses an initial anodization step, followed by removing the oxide layer and a subsequent second anodization step over the nanotextured substrate.

On the other hand, the oxide’s microstructure and degree of crystallization are controlled by post-treatments annealing procedures in air or under a suitable gas atmosphere. The composition of the oxide surface can be also be tuned in this step. The anodization of Cu in alkaline solutions, for instance, leads to the formation of copper hydroxides that can be converted into CuO or Cu_2_O species after an annealing step [42]. Kang et al. [51] utilized a CO atmosphere to convert anodized WO_3_ into WC for a cathode in PEC applications.

The morphologies, compositions, and microstructures of the anodic films are usually analyzed by field emission scanning electron microscopy (FESEM), X-ray diffraction (XRD), transmission electron microscopy (TEM), and X-ray photoelectron microscopy (XPS) techniques. The electronic properties are commonly evaluated by diffuse reflectance spectroscopy (DRS) and electrochemical impedance spectroscopy (EIS).

The usual procedure adopted to form the nanotube arrays of TiO_2_ for energy applications is the conventional potentiostatic anodization of high-purity Ti foils in organic media containing fluoride ions plus a small amount of water. NH_4_F and H_2_O dissolved in ethylene glycol (EG) have been the preferred formulae for electrolyte composition [52,53,54,55,56,57]. Other studies substituted EG with glycerol and HF by NaF [11,58]. Aqueous solutions, such as CH_3_COOH + HF [51] and H_3_PO_4_ + NH_4_F [59], were also reported. It is possible to obtain nanotube arrays with a diameter ranging from 40 to 160 nm while applying potentials in the 10–60 V range for anodization times varying from 20 min to 24 h [29,49,59,60]. The high surface area provided a larger number of active sites for the photo and electrochemical processes. In some cases, two-step anodization was carried out to obtain a high-ordered nanotube array [18,53,61]. The syntheses were usually performed at room temperature, but an exception includes the anodization performed at 55 °C for the fabrication of TiO_2_-NT bioanodes in microbial fuel cell devices [11,62].

In those applications requiring a transparent conductive substrate, like in solar cells, the TiO_2_-NT membrane was detached from the Ti metal after anodizing via ultrasonication and transferred to the fluorine-doped Tin Oxide (FTO) glass, using a TiO_2_ paste to binding the layers [18,53].

Figure 1 depicts a surface micrograph obtained by FESEM of a TiO_2_-NT sample prepared via anodization of Ti foil at 25 V for 90 min in ethylene glycol containing 10% wt. water and 0.75% wt. NH_4_F, after annealing at 450 °C for 2 h. The nanotubes exhibited a pore diameter in the 50–70 nm range with approximately 10 nm of wall thickness under this condition.

Other TiO_2_ morphologies, such as nanopores [27,63] and microcones [64], can also be obtained by changing the electrolyte and anodizing conditions. To produce nanoporous TiO_2_ for an anode of a Li-ion battery, Rahman et al. [27] annealed Ti foil under Ar before anodization and combined potentiodynamic/potentiostatic-anodization methods. The anodization was carried out in a (NH_4_)_2_SO_4_ + NH_4_F + EG solution. Regarding solar cell applications, Chen et al. [63] opted for a commercial bath (Titan-color, Poligrat) maintained at 30 °C for Ti anodization at a controlled voltage (10–80 V) for 5 min. The authors also introduced H_2_O_2_ etching after anodization to produce a nanoporous TiO_2_ layer. On the other hand, Hong et al. [64] employed an H_2_SO_4_ solution at 80 °C to form TiO_2_ microcone arrays for photoanodes of a fiber dye-sensitized solar cell (FDSSC). The anodization of the Ti mesh substrate was performed at 80 V for 3 h. Different morphologies can be produced depending on the type of energy device type setting up the experimental conditions.

Independently of the morphology, TiO_2_-NTs are amorphous after anodization and need a heat-treatment step to crystallize the oxide film. The anatase phase is the preferred crystallographic phase for photocatalytic processes [1]. The annealing was usually performed at 450–600 °C under atmospheric air for 1–3 h [58,65]. Temperatures higher than 600 °C tend to destroy the nanotube architecture [65].

To produce AAO membranes as a template or scaffold, anodization of high-purity Al foils was performed in dilute oxalic acid aqueous solution at 40–60 V in low-temperatures (6–15 °C) [26,66]. Less usual for AAO fabrication, an electrolyte containing a mixture of EG + H_3_PO_4_ + H_2_O was also reported [50]. Before the anodization, annealing and polished procedures were utilized to obtain a smooth and stress-free aluminum surface [25,66,67]. Two-step anodization was also employed to favor the formation of highly ordered nanopores [2,68]. For electrodeposition-channel-guided templates of other metals, the AAO layer was removed after the anodization, usually with H_3_PO_4_ solution, also applied to widen the pores [2,25]. The removal of the aluminum substrate after the anodization, when needed, was typically carried out by immersion in a CuCl_2_ + HCl solution [66] or HgCl_2_ [50]. A subsequent metal deposition over one side of AAO is required as a conductive layer to perform the electrodeposition.

Nanoporous WO_3_ photoanodes were produced from anodization of high-purity W foils via potentiostatic [20,51] or pulsed methods [21]. The applied potential varied from 40 to 50 V, and the anodization time varied from 30 min to 20 h. Different electrolytes were employed: glycerol + NH_4_F aqueous solution [20]; (NH_4_)_2_SO_4_ + NH_4_F solution [21]; H_2_SO_4_ + NaF solution [51,69]; K_2_HPO_4_/glycerin-based solutions [70,71]. Annealing for oxide crystallization was performed at 400–500 °C for 2–3 h [20,21]. On the other hand, the fabrication of nanosheets arrays of WO_3_ for LIB applications was performed via a two-anodization method in an H_2_SO_4_ solution [72]. Each potentiostatic anodization was carried at 15 V for 2 h and 24 h. The WO_3_ layer formed in the first step was removed by ultrasonication in water. No heat treatment was required in this case.

Nanostructured Zn/ZnO for PV and PEC applications was obtained in bicarbonate media under low voltage (5–10 V) for shorter times (10–30 min) [73,74]. Heat treatment is required to crystallize the oxide (300 °C for 1 h) [75]. Zn/ZnO-hexagonal pyramid array for a Zn-ion battery anode was created by Kim et al. [76] by applying pulsed-galvanostatic anodization in an aqueous solution containing NH_4_Cl and H_2_O_2_. The main parameter used to control the Zn periodic anodizing process was the time ratio between current-on/current-off with a continuous period of 6 s. The anodization was performed for 1 min. No further annealing was reported.

Regarding PEC systems, the anodization of iron substrates was performed in EG + water + fluoride ions via potentiostatic (50 V, 15 min) [77] or pulsed [22] methods. Fe_2_O_3_ film annealing was carried out at 500 °C for 1 h in an Ar atmosphere to form the hematite phase [77].

The synthetic routes for the fabrication of copper oxides/hydroxides depend on the final application. For PEC water-splitting, the galvanostatic anodizations of Cu foil [24] and Cu-sputtered FTO [23] were performed using high-purity copper precursors in NaOH and KOH solutions, respectively, at room temperature for 3 and 20 min. The heat treatment was carried out in an Ar atmosphere at 550–600 °C for 4 h [23,24]. For a supercapacitor device, Cu(OH)_2_ nanorods were prepared by galvanostatic anodization of Cu substrate in NaOH solution at room temperature for 1800 s [78]. In this case, no further annealing was required. As an anode for LIB, a sequence of galvanostatic oxidations (10 s and 300 s) of a Cu foil intercalated with a reduction step (20 s) in NaOH solution was performed to produce Cu(OH)_2_ nanowires [79]. After heating at 250 °C for 1 h, Cu(OH)_2_ was converted into Cu_x_O nanowires.

For fuel cell applications, two-step anodization of zirconium was conducted in a (NH_4_)_2_SO_4_ + NH_4_F solution to obtain a nanotubular ZrO_2_ film for use as a solid electrolyte. The high-purity Zr foil was anodized at 10–20 V for 5–30 min [47]. No heat treatment after Zr anodization was reported. A NiO mesoporous structure was obtained for methanol oxidation via Ni anodization in NH_4_F + H_3_PO_4_ aqueous solution at 3.5 V for 5 min and annealing in air at 400 °C for 20 min [80]. On the other hand, for a supercapacitor device, Ni foil’s anodization for the formation of NiO nanopetals was carried out in oxalic acid at 50 V and −10 °C for 10 min, followed by calcination in air at 400–550 °C for 45 min.

For use as electrodes in batteries, the mesoporous SnO was obtained from the anodization of tin foil in oxalic acid at 10 V for 20 min. After being soaked in water at room temperature for times varying from 2 to 168 h, the anodized samples were heated at different temperatures (40–100 °C) for 2 h. The nanoporous Nb_2_O_5_ was prepared from anodization of high-purity Nb foil at 14 V in NH_4_F + EG electrolyte at 50 °C for 1.5 h. The heat treatment was carried out at 450 °C and 600 °C under Ar/H_2_ for 1 h. The anodization of high purity Ta for the formation of nanoporous-Ta_2_O_5_ films was performed in EG + NH_4_F electrolyte at 50 °C by applying 15 V for 2 h. The anodized film was further annealed at 450 °C under an Ar atmosphere for 2 h.

Depending on the desired properties of the resulting material, surface modification steps are needed, such as the deposition of nanoparticles [25,29,55,81], additional layers [23,82,83], or even the functionalization of the oxide surface [51,70]. The approaches adopted will be described in the next sections, according to the material’s application and required properties.

## 3. Photovoltaic Devices for Energy Conversion: Solar Cells

The most common way to convert solar energy into usable energy is by using photoelectrochemical devices, such as photovoltaic (PV) solar cells. A simple solar cell consists of a *p-n* diode. When a photon with energy higher than the material’s bandgap is absorbed, charge carriers (holes and electrons) are generated, separated, and collected at the respective electrodes, establishing a potential difference in the *p-n* junction [84,85]. A solar cell’s efficiency depends on the materials and architectures applied to minimize reflection loss and electron/hole recombination during these processes. The different categories of PV cells can be classified according to their operational functions, compositions, fabrication methods, and component materials. The commercial PV devices widely used in the market are silicon-based photovoltaic systems, with relatively high efficiencies and stability [85,86], and those based on thin-film architectures, such as CdTe and copper indium gallium di-selenide (CIGS) solar cells [87].

Currently, the demand for high-performance devices using low-cost and less-toxic materials lead to the development of other promising types, including the dye-sensitized solar cell (DSSC) [63,64], the organic photovoltaic (OPV) cell [88], and the emerging technologies quantum dot solar cell (QDSC) [89] and perovskite solar cell (PSC) [26]. Among them, DSSCs have been attracted a more significant number of studies in the field (Table 1) due to their low cost, non-toxic properties, and simple manufacturing process, which make them eligible for industrial use. Despite the low efficiency compared with Si solar cells, its low cost and simplicity turn the DSSC into a cost-effective device.

Unlike solar cells based on *p-n* diodes, a typical DSSC works as an electrochemical system, consisting of a photoanode, a counter electrode, and a liquid electrolyte. The *n-*type semiconductor most explored as the photoanode is TiO_2_ due to its electronic properties and high corrosion resistance [9,52]. This oxide can be fabricated using several chemical methods. As an alternative solution, titanium anodization is emerging because of its simplicity and the viability of obtaining nanotubular TiO_2_ structures with precise control of the length and diameter [9,52,53].

In this sense, most studies describing the use of anodic oxides in DSSC rely on TiO_2_-NT-based photoanodes. On the other hand, some studies report the use of AAO as a template for nanostructure growth or deposition masks [25,66]. In this case, the anodic oxide is not a component of the cells, but it is used to synthesize the solar cells’ raw materials. The regular nanoporous structures with high pore density tuned by anodization parameters make it an ideal model for assisted-growth arrays with nanoarchitecture design [2]. Hence, we will discuss some recent studies involving anodic oxides in solar cells with a focus on TiO_2_-NT in DSSC, and we will present other functionalities of these anodic oxides in silicon solar cell, PSC, and OPV.

### 3.1. Dye-Sensitized Solar Cells (DSSCs)

In most DSSC devices, the photoanode is composed of a transparent conductive oxide substrate and a nanocrystalline metal oxide layer with a high dye-sensitive surface area [18]. The oxide layer can be a mesoporous film, or TiO_2_ nanoparticles (TiO_2_-NP) impregnated with the dye, usually ruthenium complexes [84]. The electrons generated in the dye interphase are injected into the metal oxide and collected in the conductive substrate. Platinum is commonly used as the cathode, with relatively high performance, but its high cost has been stimulating the study of alternative materials. The two electrodes are connected externally so that the current can flow. Between the electrodes, a non-aqueous redox electrolyte [18] or an aqueous electrolyte (usually iodine media) [84] completes the circuit. The light source can be irradiated from the front side of the DSSC (from photoanode to cathode) or vice-versa [18,49]. Backside illumination is used when the conductive substrate is opaque [63]. The overall efficiency of a DSSC depends on each constituent’s optimization and operation, particularly the processes occurring in the semiconductor film.

Despite the larger surface area of TiO_2_-NPs, its random orientation, surface defects, numerous grain limits, and electron diffusion through the irregular pathway can affect electron transport and increase the charge recombination rate [18]. In this sense, the advantage of anodic-oxide-based photoanodes is the unidirectional orientation of TiO_2_-NT, which improves the electron transport efficiency along the channel [52]. Additionally, the large specific surface area per volume tuned by decreasing nanotube diameter and increasing the nanotube length can favor the diffusion and the adsorption/desorption of species [9].

For instance, to evaluate the effect of nanotube length in the performance of a DSSC, Cheong et al. [49] prepared TiO_2_–NT layers with different thicknesses (from ≈6 to 32 µm) controlled by anodizing time using one- or two-step anodization. The authors observed that the larger the NT length, the higher the efficiency, due to the improvements in dye adsorption, electron transport, and light-harvesting ability. However, after exceeding a particular length value, the overall efficiency decreased, which authors attributed to a lower charge collection at the thicker films (NT length > 18 µm).

Despite the advantages, the anodized films also present limitations. Since the nanotubes are grown in Ti opaque substrate, the light irradiation in the DSSC device must be originated from the backside. Under this configuration, the cathode and the electrolyte can absorb part of the radiation, decreasing the overall efficiency [90]. Another disadvantage is that the annealing step can enlarge the barrier layer in the metal/oxide interface, increasing the photoanode’s internal resistance [90].

To solve both NT and NP forms’ limitations, researchers are building photoanodes with hybrid structures containing TiO_2_-NP and TiO_2_-NT multilayers and using FTO as conducting substrate [18,53,90,91]. These hybrid structures maintain light absorption efficiency by NPs with larger surface areas and the best unidirectional electron transport from NTs. In some architectures, a layer of TiO_2_-NP paste is adding between the FTO and nanotube membrane using the doctor-blade method to promote adhesion between these layers [90,91]. In other architectures, the TiO_2_-NP is deposited over TiO_2_-NT in the nanotube/electrolyte interface by electrophoresis [18] or dip-coating [64] methods to increase the dye absorption. In general, these approaches resulted in the improvement of photoconversion, electron transport, and electrolyte diffusion.

The reported technical issues with binding agents and low DSSC performance in NT- and NP-based material reveal that there are still technical gaps to improve these devices’ performance using anodic oxides. By applying selective etching to form a both-ends-opened TiO_2_-NT and sandwich it between TiO_2_-NP layers, Hossain et al. [53] observed a remarkable overall efficiency of 8.56% in DSSC performance. As a reference, the authors utilized a similar system with FTO/TiO_2_-NP photoanodes with different thickness controlled by the number of TiO_2_-NP layers, where the overall efficiency varies from 2.98% to 5.84%. The significant improvement in NT and NP’s hybrid structure was ascribed to superior light scattering properties, which increased the photo-generated electrons’ lifetime, facilitating the charge transfer through the highly ordered nanotubes.

The addition of ZnO-NP in hybrid TiO_2_-NT/TiO_2_-NP anodes was also explored by Chamanzadeh and co-authors [81]. According to them, ZnO presents a similar electronic structure than TiO_2_, but with faster electron mobility and lower photoinactivation. An increase of 43% in the efficiency was observed compared with FTO/TiO_2_-NT/TiO_2_-NPs photoanodes, attributed to reducing the electron recombination. However, larger amounts of ZnO-NP can lead to the formation of dye-Zn ion complexes, which affects the rate of electron injection into the TiO_2_ conduction band diminishing the current density [81].

Table 1 depicts the anodic oxides architecture used in DSSC devices and performance parameters: the short-circuit current densities (*J_SC_*), i.e., current density value when *V* = 0; open-circuit voltage (*V_oc_*); and overall solar cell efficiency (*η*). Due to the different conditions of experiments, these values should not be directly compared to each other. Still, it can be used as a guide in the study of materials of similar architecture.

The use of hierarchical nanostructures of the core-shell type also allows one to optimize the electron transport provided by the nanowire core and obtain a high surface area, chemical stability, and compatibility with several dyes and electrolytes [75]. Compared with conventional DSSC devices, Millers et al. [75] observed a 29% higher conversion efficiency when a ZnO@TiO_2_ photoelectrode was employed. In this case, the anodization technique was used to produce ZnO nanowires using Zn foil as a precursor. The ZnO nanowire was obtained from Zn anodization and further modified by a TiO_2_ nanofilm shell via a solvothermal method.

A variation of DSSCs composed of flexible wire photoanodes is called the flexible fiber-type dye-sensitized solar cell (FDSSC), used as a power source for soft electronics. This device exhibits a high capacity for converting photons into electricity, even under low light intensity. For this, the photoanodes should exhibit excellent flexibility, high dye adsorption, superior harvesting, and efficient carrier transfer [17,64]. Until the moment, there are few studies published recently involving FDSSCs and anodic oxides. In those studies, the anodization was performed in Ti wires as substrates. The authors aimed to optimize anodizing conditions to tune the morphology of TiO_2_-NT, such as anodizing time, potential [17], and electrolyte concentration [64], to deposit TiO_2_-NPs in a second step. Despite being in the development stage, these studies presented potential results and opened new possibilities for anodized TiO_2_ nanostructures in FDSSC devices.

Figure 2 illustrates some of the photoanode architectures built by Xiao and Lin [17] and the corresponding linear stripping voltammetric (LSV) curves. They applied different anodizing times and potential. The optimized condition was obtained for TiO_2_-NT (TNT) anodized from Ti wires at 60 V in 0.3 wt.% NH_4_F + 8 wt.% H_2_O in EG electrolyte for 6 h and annealed in air at 500 °C for 1 h (Figure 2A,B). The TNT was coated with TiO_2_-NP (TNP) by dip-coating technique (Figure 2C,D) at a withdrawing rate of 40 cm.min^−1^. To assemble the FDSSC, the authors immersed the photoanode and the counter electrode (Pt) in 0.3 mM N719 dye solution in a glass capillary (Figure 2E,F) and in a plastic tube (Figure 2H) and sealed. The experiments were performed under solar simulated light irradiation. Due to the higher light absorption of the plastic tube than that of the glass tube, the performance decrease. A J_SC_ retention of 84% was achieved for the flexible FDSSC bent 10 times than that without bending. According to the authors, these TiO_2_ composites on Ti wires open new insights for FDSSC. Still, the plastic tube’s flexibility should be improved to avoid the reduction of the light transmittance after bending several times.

### 3.2. Other Functionalities of Anodic Oxides in Silicon, PSC, and OPV Solar Cells

Among solar cells based on silicon technology, recent studies involving the anodic oxides aimed to improve the PV performance by enhancing the optical and plasmon absorption properties of metallic nanoparticles in the antireflective layers [25,66,92]. In some investigations, the anodic aluminum oxide (AAO) masks worked as a template for the deposition of plasmonic nanoparticles such as silver, indium, and gold [25,66]. The surface plasmon resonance (SPR) effect exhibited by the metallic NPs is highly influenced by metal’s nanoscale properties, such as the size, the shape of the particles, and the supporting material. The advantage of using AAO templates to synthesize these metallic NPs is that the highly ordered porous structure is easy to control and produce [66].

Ho et al. [66] prepared a thin AAO layer (≈200 nm thickness) from Al anodization with a pore diameter in the 80–100 nm range for Ag-NP deposition over a TiO_2_ spacer layer. In another study [25], these authors fabricated AAO masks with a 700 nm thickness and pore diameter of 100–110 nm to deposit Ag, In, and Al nanoparticles over the TiO_2_ spacer layer. According to the authors [25,66], this approach enabled the fabrication of plasmonic Si solar cells with an overall efficiency higher than those produced without NPs. The better performance was attributed to the plasmonic effects of the metallic NPs.

Cao et al. [92] applied the anodization technique to produce a honeycomb-textured Ti substrate to deposit Cu, Au, and Ag as back reflectors. Like the AAO template, the Ti substrate is not part of the active layer but acted as support for metallic NPs. The Ti foil was anodized in EG, fluoride, and water media to form the nanotube arrays. Then the oxide was dissolved via ultrasonication in water remaining only the nanotextured Ti substrate.

The perovskite-type cells (PSC) are a new generation of solar cells known for their high efficiencies, low-cost, flexibility, and semitransparency, which expand their fields of application [26,93]. These solar cells are composed of an organometal halide perovskite with chemical formula ABX_3_ where A = methylammonium or formamidinium; B = Pb or Sn; and X = I, Cl, or Br [26], but the most common material is Pb [26,94]. Current studies involving PSC and anodic oxides applied the AAO scaffold for the perovskite deposition since the alumina allows the control of the perovskite layer’s spatial distribution and volume, which influences the optical properties, such as transparency and color neutrality [26,94]. Unlike the templates, the AAO scaffolds are not dissolved after deposition, remaining in the structure.

Some studies also report using anodized TiO_2_ in PSC and hybrid OPV/PSC devices [95,96]. In these cases, the Ti/TiO_2_ substrates were incorporated into the active layer to enhance charge conduction. Yang et al. [95] observed a substantial increase in photocurrent density than a PSC without TiO_2_-NT, which authors ascribed to enhancing the charge conduction and sunlight harvesting.

Despite the PSC potential, the Pb toxicity and poor photostability challenge its extensive applications and limit device sustainability. Therefore, the scientific community has been searching for less toxic perovskite structures or Pb-free materials, such as Sn, Ge, Bi, and Sb [97]. However, these new materials are still not close to APbI_3_ solar cells [97]. No studies were found using anodic oxides in a Pb-free perovskite device, which may be an opportunity to be explored in further studies in this field.

In the organic photovoltaic devices, the film’s transparency and optical haze are essential factors to consider, since they affect the light behavior. Kang et al. [88] proposed a novel treatment for obtained a thin and transparent haze film composed of self-aggregating alumina nanowires from anodized AAO membranes. By connecting three different types of haze films to the polymer/fullerene OPV device’s front surface, a relative improvement of 5 to 10% in energy conversion efficiency was obtained. This result was attributed to the long optical path’s length in the device’s active layer by the passage of scattered light through the haze film. Table 2 depicts the most common applications of anodic oxides in silicon, PSC, and OPV devices.

The study and development of sustainable energy conversion systems such as PV solar cells are fundamental regarding environmental demands. As observed, nanostructured anodic oxides can be used as a component in several types of PV-solar cells. Their stability and versatility allow different functionalities (photoanode, template, scaffold, haze film). In this sense, these materials can help achieve high photoconversion efficiency, improve device performance, scale-up to industrial applications, and reduce costs.

## 4. Photoelectrochemical Devices for H_2_ Production: PEC Water-Splitting Cells

Due to its high gravimetric energy density, abundance, and no effect on greenhouse gas emissions [98], hydrogen is a potential alternative to the use of fossil fuels. This gas can be generated by several processes [98], but one of the cleanest and sustainable technologies is its production from the photoelectrochemical (PEC) water splitting. Similarly to the water electrolysis process but utilizing photo processes as an additional drive force, the PEC devices can use solar and electric energy for splitting the water molecules into hydrogen and oxygen molecules. Since these cells are composed of electrodes immersed in an aqueous electrolyte, where the products are collected separately, the hydrogen can be stored and used for other applications, such as fuel in nonpolluting vehicles or reagents for the chemical industry, or it can be used to generate electricity in fuel cells.

Despite this green technology’s potential, the limited performance and high cost of the raw materials postponed the practical applications of PEC devices for hydrogen production from water splitting [51]. The overall efficiency of this process depends on electrode properties and device configuration. To make this technology commercially viable, it is necessary to scale-up the fabrication process, reduce the costs, and improve the cell’s performance by enhancing the stability and durability of the high surface area electrodes [99,100].

In the water-splitting process from a PEC device, charge carriers are photo-generated and driven to the electrode/electrolyte interface to produce gaseous O_2_ and H_2_. The oxygen evolution reaction (OER) occurs in the anode interface, where the oxygen of the water molecule is oxidized to O_2_ [10]. For this, these electrodes are usually composed of *n*-type semiconductors with photoactive properties [10]. In the cathode, where the reduction of protons to H_2_ leads to hydrogen evolution reaction (HER), *p-*type semiconductors have been applied as alternatives to the noble metals [20,101]. The compartments containing the anolyte and catholyte, where the OER and HER occur, as described by Equations (1) and (2), respectively, are usually separated by a proton-conductive polymeric membrane to avoid back reactions and separation steps.
(1)$$\mathrm{Photoanode}:\;H_{2}O_{\;(l)}\;+\;h\nu \;\to \;½\;O_{2\;(g)}\;+\;2\;H^{+}{}_{\;(\mathrm{aq})}\;+\;2e^{-}$$
Cathode: 2 H^+^_(aq)_ + 2e^−^ → H_2 (g)_(2)

Several factors must be considered: the mobility of charges must be adequate to assure a high efficiency; the photo-generated electrons at the photoanode cannot recombine with protons; a thicker layer should be able to absorb a larger quantity of light; a high surface area with active sites must be available [10,54]. Besides photoactivity efficiency, the kinetics aspects must also be taking into account cause the OER in the anode is, generally, the limiting step of the mechanism reaction [51]. For these reasons, the most strategies for improving PEC efficiency lie in designing and investigating the photoanodes properties instead of the cathodes.

Due to the chemical stability, metallic oxides produced by several methods have been explored as photoanodes. Regarding those prepared via anodization, the technique provides the advantage of the fabrication of high ordered nanostructure with a large surface area ideal for charge transfer and transport and the absorption and desorption of reactive species [92]. According to Chiarello et al. [61], the ordered structure can confer to the oxide photonic crystal properties, allowing confine and control photons, increasing light-harvesting and absorption efficiency.

The recent papers describing the water-splitting devices using anodic oxides mainly explored photoanodes based on TiO_2_-NT [10,19,54,82,102,103,104], WO_x_ [20,21,51,69,70,71,101], Fe_2_O_3_ [22,77], and ZnO [73,74]. These studies focus on improving the photocurrent density and overall PEC efficiency by developing electrodes with specific bandgap properties, crystallinity, and morphology.

Some studies report using the anodization technique to produce TiO_2_-NT films without surface modification or doping, except for the annealing [10,19,54]. In these cases, the photoanodes were obtained via anodization of titanium in the form of foils [10], mesh [54], or a web of microfibers [19]. Using a compact PEC water-splitting device, Ampelli et al. [10] tested TiO_2_-NT thin films with different thicknesses controlled by anodizing time (30 min to 5 h). The results showed that the photocurrent response decreased with the enlargement of the film. An optimal efficiency was obtained for the thinner film, produced using a short anodizing time, even in the absence of an external bias. The authors suggested that the enhancement of visible light absorption was higher for the thicker film. Still, this effect was negatively counterbalanced by the increase of charge recombination with the enlargement oxide layer [10]. In other investigations, Saboo et al. [54] built a 3D hierarchical structure in the photoanode by anodizing a Ti mesh using different potentials and times to favor the mass transport water-splitting process. The results showed that the inner diameter and the length of nanotubes influenced the PEC cell’s performances. The thinner film produced at short anodization time had good efficiency, a similar effect observed for Ampeli and co-workers [10]. Stoll et al. [19] applied Ti webs as a substrate for TiO_2_-NT and evaluated the H_2_ production in a PEC device design to operate in both liquid and gas phases. The photoanode exhibited superior performance in comparison with the conventional powder semiconductors on a carbon substrate. Figure 3 depicts the PEC cell designed for H_2_ production from water splitting. The cell was composed of a membrane electrode assembly containing the TiO_2_-NT photoanode, a Pt/C cathode, a Pt/C reference electrode, and a proton conducting polymeric electrolyte membrane, which acted as compact reactor for water splitting and as gas separator [19].

Due to its wide bandgap, TiO_2_ is photo-responsive under UV irradiation [105]. However, only 5% of solar radiation comprises the UV region of the electromagnetic spectrum [105]. Therefore, to enhance photoactivity, TiO_2_ has been modified with different materials to change its band structure, sensitize it to visible light, and decrease the charge carriers’ recombination. The most common strategies to alter the electronic properties of the semiconductor involve the addition of metals [102,104], semi-metals [103], metal oxides [82], and non-metals [105] as a dopant or deposits over the oxide film. Anodic oxides can fill this gap since it is possible to change the electrolyte composition by adding specific elements in the anodic material during the oxide formation.

Nickel is one of the metals used to improve light absorption and separation of photo-excited electron-hole pairs. Its oxide, NiO, is a *p*-type semiconductor with a high concentration of holes [82], commonly coupled to *n*-type semiconductors as a co-catalyst. Compared with a bare TiO_2_-NT anode, Rasheed et al. [82] increased the photocurrent densities in pure water by depositing a NiO layer on TiO_2_-NT. For this, the nanotube film was soaking in a solution containing a Ni salt and calcinated at 340 °C for 2 h. According to the authors, the *p*–*n* junction at NiO/TiO_2_ interface promoted interfacial carriers’ transfer and the separation of e^−^/h^+^ pairs, enhancing PEC’s efficiency.

Instead of producing a new phase containing Ni, Dong et al. [102] anodized a Ti alloy containing 1 wt.% of Ni to dope TiO_2_-NT. The photoconversion efficiency improved 3.35 times than the photoconversion efficiency of undoped TiO_2_. In another study, the same authors tested the effect of Si doping by producing TiO_2_-NT from a Ti alloy containing 5 wt.% Si [103]. In this case, the photoconversion efficiency increased about 2.7 times when compared with undoped TiO_2_. The authors [102,103] attributed both studies’ results (with Ni doping and Si doping) to improve light absorption and electron-hole pair separation promoted by the dopant addition.

Another strategy to improve the absorption range under visible-light irradiation was proposed by Momeni et al. [104], which sensitized TiO_2_-NT with W and Cu. Compared with a bare TiO_2_-NT, the co-sensitized TiO_2_-NT exhibited a broad absorption range under visible light, enhancing photocatalytic performance. A heterostructure combining quantum dots (QD) and TiO_2_-NT also demonstrated significant results considering visible light sensitization [105].

Due to its high stability, resistance to photocorrosion, and suitable band gap value to absorption in visible light, WO_3_ is also an attractive candidate for PEC systems [20,21,71]. A comparison between a commercial WO_3_/W electrode and an anodized WO_3_-NT/W performed by Li et al. [20] revealed that the oxide produced via anodization of W foil exhibited a stable and higher photoactivity for visible-light-driven water splitting. This result was attributed to the fast electron-hole separation rate in the nanostructured oxide, strong interaction between metallic W and oxide layer, and the high crystallinity of WO_3_. Das et al. [71] fabricated a bilayered WO_3_ film composed of a porous layer and a thicker barrier layer that exhibited promising water splitting performance. According to the authors, this architecture promoted significant and efficient charge transfer/transport events at the interface, resulting in improved PEC performance. Lu et al. [21] also observed excellent stability and high photoconversion efficiency using the WO_3_ nanoporous electrode produced via pulsed anodization technique. Other preliminary studies reported the effect of anodization conditions on the photoactivity of WO_3_ by investigating the influence of the temperature [69,101], the substrate [101], and doping [70] on the properties of the oxide.

With a suitable bandgap to absorb visible-light irradiation, the use of hematite (Fe_2_O_3_) obtained via iron anodization was also explored for water splitting applications despite its low conductivity [22,77]. To improve PEC water-splitting performance, Lv et al. [22] doped hematite by adding Sn^2+^ ions in the electrolyte during the synthesis to enhance the oxide film’s conductivity. The authors observed an increase of IPCE (incident photon-to-current efficiency) from 2.4% to 4% under 400 nm irradiation when the dopant was added into hematite. Results also showed that IPCE obtained under the same conditions improved 6-fold with the deposition of Co phosphate (Co-Pi) co-catalyst over the Sn-doped Fe_2_O_3_ photoanode, which was ascribed to the hole transfer efficiency.

Besides hematite, Zn anodization also demonstrated potential for ZnO nanowires’ production as photoanodes for PEC water splitting [73,74]. With a wide bandgap (3.37 eV) and absorption in the UV region, ZnO exhibits high electron mobility, from 10 to 100 times higher than TiO_2_ [74]. Other properties that make this n-type semiconductor attractive for PV–PEC processes are its high photostability, thermal stability, low-cost, low-toxicity, and biodegradability [74]. However, these studies are in the early stage, focusing on investigating the correlation between synthetic parameters, experimental conditions, photocurrent response, and photostability.

Table 3 depicts photoanodes’ architecture based on anodic oxides and maximum photoconversion efficiency values measured under indicated conditions, grouped according to the type of the irradiation source: simulated solar light or monochromatic irradiation (visible light and UV-A). It is important to stress that a direct comparison between efficiencies is not suitable since the formulae used to determine the efficiency can differ from one study to another. These values were also obtained under different cells, electrolytes, light irradiation, and experimental conditions. Despite this, data are compiled as an illustration of the different architectures applied, and it can work as a guide for the design of new experiments. For photocurrent densities values, see the original papers. A PEC cell’s efficiency depends on applied bias and wavelength of the light irradiation source so that the maximum efficiency expresses the optimized condition. This efficiency can be determined according to the following models: the solar-to-hydrogen conversion efficiency (STH), applied bias photon-to-current efficiency (ABPE), quantum efficiency (QE), incident photon-to-current efficiency (IPCE), and absorbed photon-to-current efficiency (APCE), this latter is also known as internal quantum efficiency [10,20,99,106].

The use of AAO as a template to grow nanostructures inside the pores is also being explored in the synthesis of different photoanodes, such as Cu_2_ZnSnS_4_ (CZTS) [68], TiO_2_-doped WO_3_ [107], and organic [108] semiconductors. Aiming a large-scale production of PEC devices, Kim et al. [68] proposed a synthetic route to fabricate vertical arrays of 1D CZTS nanostructures into nanopores of AAO membranes, including the immersion of the AAO templates into a CZTS solution precursor following by an annealing process. Bendova et al. [107] used AAO templates to produce TiO_2_−doped WO_3_ nanorods photoanode by anodizing W/Ti layers through the alumina nanopores. Guo et al. [108] also used AAO as templates to produce graphitic carbon nitride nanorods by a thermal condensation process from a cyanamide precursor into alumina nanopores.

Considering the cathode where HER occurs, few papers that have been published recently described any application of nanostructured anodic oxides as the cathode [51,109]. Kang et al. [51], for instance, anodized W foils to produce WO_3_ and convert them to crystalline tungsten carbide after annealing in CO atmosphere. The resulting WC was tested as the cathode in a PEC water-splitting device. The PEC device was coupled to a DSSC based on TiO_2_/FTO photoanodes. The performance of the WC cathode was compared with a Pt cathode. The results revealed a significant increase of ≈25% in energy conversion efficiency when Pt cathode was replaced by WC film [51]. Despite not being tested in a practical PEC device, Jian et al. [99] anodized Mo foils to produce molybdenum oxide rich in oxygen vacancies aiming to enhance HER in the cathode from alkaline solutions. The MoO_x_/Mo electrode exhibited high activity and excellent stability in continuous operation in a KOH solution.

With bandgap suitable for absorption in the visible-light region [23,24,110,111], the photoactivity of the *p*-type semiconductors based on copper oxides has also been explored as photocathodes in PEC water-splitting applications. In these cases, the cathode instead of the anode is irradiated. The abundance, non-toxicity, and scalability of the anodization method have motivated these studies. However, the corrosion of copper oxides promoted by photo and electrochemical processes is still a drawback for PEC processes’ practical use. Luo et al. [23] explored CuO–Cu_2_O nanowires as photocathode for hydrogen generation, fabricated via anodization of Cu film sputtered on FTO substrates. Therefore, to improve the stability of these materials, the authors prepared a protective layer of Al-doped ZnO and TiO_2_ decorated with a RuO_x_ catalyst and inhibited the contact of the exposed parts of the Cu metal substrate with the electrolyte creating a blocking layer by passivation. The resulting nanostructured exhibited an increase of 25% in photoactivity under visible light irradiation than planar electrodes of similar architecture. Another approach applied for improving the stability of Cu_2_O nanowires for water splitting was proposed by Shi et al. [24], which embedded the anodized electrodes into glucose solution to form a C-coated Cu_2_O photocathode after annealing. According to the authors, the enhanced stability decreased the photo-corrosion rate and accelerated the charge transfer processes at the electrode/electrolyte interface, increasing the photocurrent response.

The anodic oxides demonstrated the feasibility and practical utilization of the electrochemical synthesis to manufacture PEC water-splitting components for both photoanode and photocathode. However, the main contribution lies in the photoanodes architecture design. The photoactivity properties combined with aligned directional pathways for charge transfer of the nanostructured anodic oxides have improved photoconversion, stability, and PEC devices’ overall performance. As well in the PV systems, these features make this green energy conversion process more efficient, economical, and attractive for practical devices.

## 5. Electrochemical Devices for Energy Conversion: Fuel Cells

A fuel cell is a power source device that converts the chemical energy of a substance directly into electricity. These devices consider a promising sustainable technology for electric energy production since it operates without combustion cycles and produces non-toxic pollutants and non-acoustic emissions. A fuel cell is composed of an electrolyte and electrodes (anode and cathode), operating via an electrochemical reaction between the fuel (H_2_, CH_3_OH, CH_2_O_2_) and an oxidant (usually O_2_ from the air) [112,113]. Given that it is an open system, it requires a continuous supply of the reagents. Depending on the fuel, the main products are H_2_O, CO_2_, and heat.

The fuel cells are categorized according to the fuel or the electrolyte employed. Herein, we point out those fuel cells in which anodic oxides were employed as parts of its components, such as Proton-Exchange Membrane Fuel Cell or Polymer-Electrolyte Membrane Fuel Cell (PEMFC), Direct Formic Acid Fuel Cells (DFAFC), Direct Methanol Fuel Cell (DMFC), Solid Oxide Fuel Cell (SOFC) and Microbial Fuel cell (MFC). These fuel cells diverge in operating temperatures and efficiency. Still, their operation is similar: the fuel is oxidized in the anode surface and transported by an ion-conducting electrolyte until the cathode, where the oxygen reduction reaction (ORR) takes place [112].

Nowadays, fuel cells are still in the development stage. Despite their higher energy density, much work must be done to produce an efficient and economical fuel cell. The materials’ high cost is one of the main barriers to commercializing this technology [55,114]. One of the main strategies for reducing costs is to improve the electrolytes and electrodes by applying porous nanomaterials with corrosion-resistance, stability, and durability [29,84]. In this sense, nanostructured anodic oxides are potential candidates for components of the fuel cells. However, few papers were published in the literature in the last few years describing devices or practical fuel cells using the anodized oxides as some of their components. Most of these studies evaluated the potential use of the anodic oxides in the fuel cell electrodes focusing more on the material’s synthesis than the application. In these studies, anodized TiO_2_-NTs are tested as supporting materials for the electrocatalyst in the electrodes for PEMFC, DMFC, and DFAFC applications [29,30,51,55,58]. The evaluation of anodic Y-doped ZrO_2_ as the solid electrolyte for SOFC applications was also reported [47]. In MFCs, TiO_2_-NTs were used directly as bioanodes for energy production from biological fuel extracted from organic compounds [11,60,62,65].

Besides ZrO_2_ [47] and NiO [80], the anodized TiO_2_-NT is the widely explored anodic oxide in fuel cell applications. Some authors [29,55] asserts that TiO_2_-NTs play a significant role in the electro-oxidation of small organic molecules such as methanol and formic acid. However, the electrocatalytic activity of this semiconductor was not investigated in detail in these studies. Instead, the TiO_2_-NT in these applications is motivated by its high surface area and chemical stability, ideal for the catalyst. These features can enhance the fuel cell’s performance and long-term stability since the commercial carbon-based materials degraded with operation time. On the other hand, some biofuel cell applications explored the direct use of TiO_2_-NTs without any further metal deposition. In these cases, due to biocompatibility and electron transfer properties, TiO_2_ acts as the electrocatalyst for electricity generation from the biological microorganism metabolisms in MFCs.

The general aspects and the most significant results involving the anodic oxides in fuel cell applications will be discussed in the sequence according to the fuel cell type. The list of the anodic oxide materials fabricated via anodization explored in these applications are summarized in Table 4:

### 5.1. Proton-Exchange-Membrane or Polymer-Electrolyte-Membrane Fuel Cells (PEMFC)

In a PEMFC, the process of interest is the reverse reaction of the water splitting: protons from H_2_ fuel must react with O_2_ to form water molecules. From a commercial point-of-view, this device is the only fuel cell capable of generating power for practical applications. This fuel cell presents high power density and efficiency by operating at 60–120 °C temperature [74,107]. The disadvantage is the miniaturized hydrogen containers’ high cost and the materials since the acidic media requires Pt-based electrodes [55]. To reduce the materials’ costs, conductive carbon materials (nanotubes, nanofibers, or graphene) with a high surface area had been used as support material for Pt nanoparticles [30]. However, the low corrosion resistance and the aging effect on the electroactive area due to Pt aggregation cause the fuel cell’s performance degradation, reducing the PEMFC lifetime [30,55].

As an alternative, semiconducting metal oxides are considered good support material for the electrocatalyst anchoring due to their chemical stability. Among them, the low cost, eco-friendliness, high stability in both acidic and alkaline solutions, and high surface area make TiO_2_-NT a potential candidate for supporting the catalyst. Manikandan et al. [30] evaluated the electrochemical performance of TiO_2_-NT decorated with Pt as cathode for the oxygen reduction reaction (ORR) in PEMFC applications. Pt-NP were deposited on TiO_2_-NT from NaBH_4_ precursor via a chemical reduction method. The electrochemical performance of Pt/TiO_2_-NT was compared with Pt-NP deposited in the other two supports: TiO_2_ nanoparticles (Pt/TiO_2_-NP) and commercial carbon (Pt/C). The Pt/TiO_2_-NT electrode exhibited long-term stability after 10,000 cycles and higher electrocatalytic activity. The authors attributed these results to enhanced charge transfer, high corrosion resistance, and the ability of TiO_2_-NT to anchor Pt uniformly. Despite promising results, the authors did not test this material’s performance in a practical PEMFC system.

### 5.2. Direct Methanol Fuel Cell (DMFC)

Unlike PEMFC, the fuel in a DMFC is inserted in the liquid state. In this device, the methanol (CH_3_OH) is oxidized to CO_2_ on a catalyst surface in acidic or alkaline media. The high-density energy and ambient operational temperature make this fuel cell attractive. However, the slow kinetics of methanol electro-oxidation and the membranes’ low conductivity are still a challenge for improving the DMFC efficiency [29]. Additionally, the poisoning of the surface by chemisorbed intermediates such as CO can block the catalyst’s surface, generally Pt-based materials. In PEMFC applications, the strategy to overcome these problems is to substitute the electrodes for new materials with high electrochemical activity and good stability [29]. The anodic oxides have been demonstrated to be a promising alternative.

Among different alternatives to Pt-based catalyst, Nickel-based anodes are the most investigated due to high electrocatalytic performance, reduced cost, and good stability [29,80]. Using anodization, some authors prepared anodic oxides from Ni and Ti substrates for methanol oxidation, producing mesoporous NiO [80] and Ni-modified TiO_2_-NT [29]. Wang et al. [80] evaluated the performance of mesoporous NiO prepared via anodization of Ni foil. The current peak due to methanol oxidation to CO_2_ was significantly higher than other Ni-based materials in a similar electrolyte. Additionally, fast reaction kinetics was detected, ascribed by the authors to the highly porous structure with large amounts of grain boundary that facilitated electro-active material access. The authors also observed a better electro-stability than other Ni-based catalysts and no poising effect by intermediates.

Haskul et al. [29] evaluated Ni as an electrocatalyst supported in TiO_2_-NT. The nanotube array was prepared by Ti anodization, and different Ni content was further electrodeposited over the TiO_2_ film. The methanol oxidation peak was not detected using the bare TiO_2_-NT electrode, but only using the Ni/TiO_2_-NT electrode. The results showed the current densities increased with the Ni content and with electrolyte temperature; stability tests revealed that the current retained 85% of the original value after 100 cycles. According to the authors, TiO_2_-NTs with large surface areas exert a significant role in methanol oxidation and can be explored as support for the electrocatalyst in DMFC applications.

From a different point of view, Liang et al. [115] applied the anodization technique to produce hydrophobic TiO_2_ in degassing channels of the anode in a DFMC. The authors’ problem of DMFCs performance was the capillarity blockage caused by CO_2_ accumulation in anode flow channels, leading to the deterioration of the DMFC performance with time. To solve this problem, the authors proposed a novel design for the anode flow where hydrophobic TiO_2_ degassing channels remove CO_2_ before it is released to the fuel channel. The authors combined photolithography, etching, anodization, and fluorination treatments to fabricate a micro-nano TiO_2_ hierarchical structure on Ti substrates [115]. Compared with the conventional DMFC, they observed a superior performance with higher power densities at moderate methanol flow rates. According to the authors, the superhydrophobic degassing channels significantly accelerated the exhaust of CO_2_, leading to more efficient CO_2_ removal in N-spiral and improved the DMFC performance.

### 5.3. Direct Formic Acid Fuel Cells (DFAFC)

The limited compatibility of methanol with the ion-conducting membrane of the electrolyte restrain the fuel to a low concentration in a DMFC [117]. Besides, the methanol toxicity in the vapor phase limits the commercialization of DMFC technology. An alternative to methanol as liquid fuel is the formic acid, which is non-toxic in dilute solution and presenting a smaller crossover flux through the ion-conducting membrane than methanol [117]. Additionally, with a faster oxidation reaction, the formic acid can be produced from biomass or sequestrated CO_2_ [58], reducing CO_2_ in the environment. Its primary disadvantage is a volumetric energy density lower than methanol, but it can be compensated for using a high concentration of formic acid. Studies in the literature [117] reported electrocatalytic oxidation activities of DFAFCs superior to the DMFCs and, in some cases, close to H_2_-PEMFCs.

In terms of materials, Pt is also the wide electrocatalyst used in DFAFC. Still, it is often poisoned by CO intermediates’ adsorption, a similar effect observed in methanol electrooxidation in DMFCs. The indirect electrooxidation on Pt electrode proceeds according to [58]:HCOOH → CO_ad_ + H_2_O → CO_2_ + 2H^+^ + 2e^−^(3)

To promote the direct CO_2_ oxidation reducing CO formation, researchers replaced platinum with palladium [55,58,117]. Anodic oxide films’ potential as supporting material for the electrocatalyst for DFACF applications was evaluated in some studies [55,58,59]. In a preliminary study, Abraham et al. [55] electrodeposited Pd dendrites on TiO_2_-NT to evaluate its catalytic activity in formic acid electro-oxidation. The results showed that the electroactivity for formic acid oxidation enhanced when compared to Pd dendrites deposited directly on Ti metallic substrates. Other papers report the deposition of both Pd and Pt nanoparticles on TiO_2_-NT substrates using magnetron sputtering [58] and UV-light induced deposition [59] techniques. According to Li et al. [59], Pd–Pt materials exhibited the highest current density and long-term stability for formic acid oxidation, which the authors ascribed to the synergetic effect between Pd and Pt. The authors also reported a significant anti-CO poisoning ability.

Pisarek et al. [58] tested the TiO_2_-NT decorated with Pt, Pd, and Pd–Pt–NP as an anode in a DFAFC and compared the activity of these materials with the commercial catalyst—Pd/Vulcan, a carbon-based material. Similar to those produced on a carbon substrate, the voltammogram profiles indicated an excellent conductivity of TiO_2_-NTs and no barrier for electron transfer at the interface oxide/NPs. Despite the synthesized material presented a lower specific surface area than the Pd/Vulcan, its activity was significantly higher than the commercial one, as shown in Figure 4.

Figure 4 shows the results of DFAFC tests where it can be noted that the Pd/TiO_2_ leads to a maximum of specific power 70% higher than that of the commercial Pd/Vulcan [58]. The authors attributed the Pd/TiO2’s superior performance to the other materials to the metal-support interaction, the hydrophilicity of TiO_2_, the morphology, and the crystalline size, which make the material a promising anode for DFAFC.

### 5.4. Solid Oxide Fuel Cell (SOFC)

A SOFC is a high-temperature operational fuel cell (800–1000 °C) with an ion-conducting solid electrolyte between the cathode and the anode, which is the main component of this fuel cell [112,113]. The solid oxide’s high ionic conductivity requires a high-temperature operation, although some papers also described operational temperatures of about 500 °C [113]. The operation is similar to the PEMFCs. However, the SOFCs are more flexible since they also work by using hydrocarbons as fuel. The more common solid electrolyte used is composed of Y_2_O_3_-stabilized ZrO_2_ (YSZ) [47,112].

The potential use of the anodized metal oxides in SOFC applications was described in two studies. One explored ZrO_2_-NT as the solid electrolyte [47], and the other tested AAO membranes as electrocatalyst support [116]. Buyukaksoy et al. [47] evaluated the electrical properties of undoped ZrO_2_-NTs prepared via anodization technique as a solid electrolyte for SOFC applications. By tuning the experimental conditions, the authors managed to synthesize a new material with a high surface area. The activation energy and electric conductivity values at 600 °C showed consistency with other studies with YSZ-electrolyte fabricated by other methods. Despite promising results, the electrolyte was not tested in a fuel cell. Kwon et al. [116] utilized commercial anodized AAO as support for Pt electrocatalyst to improve the electrodes’ stability and reduce the fast degradation of the electrocatalyst at high temperatures (400–600 °C). The enhanced thermomechanical strength was attributed to the honeycomb-shaped nanopore structure, which also preserved the electrocatalyst’s integrity and electrical conductivity during high-temperature operation.

### 5.5. Microbial Fuel Cell (MFC)

MFCs are devices producing electricity from organic compounds by catalytic reaction of electroactive bacteria. It is considering a green technology, which can treat wastewater and produce energy simultaneously. However, its technology is limited by low power generation efficiency [60]. The differences between MFC and the conventional fuel cells are: the anode is a biofilm containing an electroactive bacteria (or a protein), and the fuel must be a fermentable substrate, such as water waste, industrial effluent, or biomass [118]. Heterotrophic bacteria can oxidize various organic molecules and produce energy, forming hydrogen, formate, acetate, and methane as byproducts. The acetate MFC has been reported as the most efficient in terms of electricity generation [118].

The main components of the MFC that limit its performance and costs are the electrodes [11]. In this sense, recent research goals in MFCs applications are to develop cost-effective electrodes and increase MFC efficiency. Besides a good electric conductivity, the anode materials’ requirements are corrosion resistance, mechanical strength, and biocompatibility. The commonly applied materials are carbon-based materials (e.g., graphite) and metallic substrates (Cu, Ni, Au, Ag, Ti, and stainless steel) [11,118]. The disadvantages of the carbonaceous materials are low electrical conductivity and low mechanical strength. On the other hand, the metallic substrates are cheap and scalable but present low corrosion resistance.

Regarding the biocompatibility and stability of TiO_2_ nanoparticles, Feng et al. [11] proposed to increase the performance of Ti electrodes by producing an anodic oxide layer of TiO_2_-NT on its surface. The experiments were performed in a dual-chambered reactor separated by a cation exchange membrane. The catholyte consisted of a K_3_[Fe(CN)_6_] solution. The anolyte (a mixture of CH_3_COONa, NH_4_Cl, NaCl, KH_2_PO_4_, K_2_HPO_4_, MgSO_4_, NaHCO_3_) was inoculated with an anodic effluent from an acetate-fed MFC reactor that had been continuously running for two years. Cyclic voltammetry of the biofilm carried out in the same medium exhibited the sigmoidal shape under acetate turnover conditions. According to the authors [11], this behavior indicated the current generation by the biofilm-associated extracellular electrons. After four days of operation, the maximum current density detected was 190 times higher than Ti bare electrodes and twice than a graphite electrode in a similar condition.

According to some authors [60], the main factors influencing the TiO_2_-NT efficiency as a bioanode in MFC are the hydroxyl groups of TiO_2_ that can enhance the bacteria’s attachment in the oxide surface, contributing directly to the transfer of redox proteins to TiO_2_. To evaluate TiO_2_-NT efficiency, Guo et al. [60] prepared two nanotube films with different diameters controlled by electrolyte composition since the other anodizing parameters were equal. The nanotubes produced in 0.2 v% HF exhibited a diameter ranging from 50–120 nm, whereas the sample anodized in the presence of 0.5 wt% NH_4_F dissolved in a 170:30 (*v*/*v*) propanetriol: H_2_O solution exhibited nanotube diameter ranging from 80–150 nm. The MFC experiment was carried out in a dual-chamber reactor containing K_3_[Fe(CN)_6_] solution, as the catholyte, and CH_3_COONa, NH_4_Cl, NaCl and MgSO_4_ solution as the anolyte, which was inoculated with the sludge supernatant from a wastewater treatment plant. The authors verified that both TiO_2_-NT electrodes demonstrated a better MFC performance when compared with a similar experiment where a carbon felt was used as the anode, as reported in the literature. However, when compared to each other, the results indicated that the nanotubes with the larger diameter and interspace improved the MFC performance. Besides the morphology, the authors attributed this good performance to the higher content of oxygen lattices and better conductivity of the TiO_2_. The authors also considered it a promising and scalable MFC bioanode due to the balance between low-cost and high performance.

Using a different approach, Ait Ali Yahia et al. [65] tested amorphous and crystalline TiO_2_-NTs substrates as the cathode for the oxygen reduction reaction (ORR), aiming to improve MFCs performance. Although the crystalline materials had shown better conductivity due to the higher oxygen vacancy density, the MFC’s highest maximum power density was achieved using the amorphous TiO_2_-NT cathode. The authors attributed these results to the lack of specific surface area for ORR in the crystalline films, and the possibility of the bulk oxygen vacancies did not correspond to surface oxygen vacancies in the reaction condition. Therefore, the authors did not consider TiO_2_-NT as an efficient catalyst for ORR. They suggest a modification by doping or sensitization to improve its efficiency as the cathode.

All these results concerning fuel cell technology reveal that each type presents advantages and drawbacks depending on the application. In general, the use of anodic oxides in their components has been evaluated as a plan to reduce cost, enhance stability, and increase the fuel cell’s durability in corrosive media and high temperatures. The anodized titanium dioxide is the most tested material, acting as support for the electrocatalyst in the conventional fuel cell or the electrocatalyst itself in the microbial fuel cell. However, many studies are required to improve these materials’ properties, and make the green fuel cell technology scalable and efficient for practical applications.

## 6. Energy Storage Devices: Supercapacitors and Batteries

The most practical form to store electric energy for consumption on demand is by using supercapacitors and batteries. These devices’ main differences are the charge/discharge mechanisms, power- and energy-densities, and long-cyclic performance [78,119]. Supercapacitors present fast charge–discharge rates and high power densities than rechargeable batteries [50].

### 6.1. Supercapacitors

Currently, supercapacitors are employed in many applications, including portable electronic devices, pacemakers, cranes, forklifts, and hybrid electric vehicles [31,78]. A capacitor consists of two conducting elements separated by a dielectric and acquires charge by applying a potential between them. On the other hand, the term supercapacitor refers to the electrochemical capacitor, which operates via a double-layer mechanism or charge transfer reactions [78,112]. The former case comprises the non-faradaic electric double-layer capacitor (EDLC). The latter point refers to the faradaic pseudocapacitor operating as a function of redox reactions. Like the other electrochemical systems, the overall performance of these devices depends on the electrode materials and their effect on the device’s properties, such as energy density, power, cyclability, and long-term stability.

In the supercapacitors, the larger the electrode material’s surface area, the higher the capacity to store charge into its surface. The common electrodes used in EDLC are carbon materials, such as activated carbon, graphene, and carbon nanotubes, with high surface areas available for the electrostatic charge storage [78,120]. Despite the higher rate, EDLC presents a lower energy density than batteries; therefore, one of the approaches for enhancing its performance is to use nanostructured carbon materials combining with oxides to provide supercapacitance [121]. Another approach is to build an asymmetric supercapacitor where one of the carbon electrodes is replaced by a pseudocapacitive material that can store more charge via fast reversible redox reactions in the electrode/electrolyte interface [78].

The electrodes consist of an active material and a current collector. Pseudocapacitive electrodes composed of conductive polymers or metal oxides (MnO_2_, NiO, V_2_O_5_, RuO_2_, Fe_2_O_3_, CoO/CoFe_2_O_4_, and ZnMn_2_O_4_ and as the active layer present excellent charge-transfer processes [48,78,120,122,123]. The conductive polymers exhibited high specific capacitance, but they are limited by poor cyclic stability [124]. On the other hand, the long-term stability of the metal oxides is determined by their low conductivity. In this sense, developing different strategies to enhance the electrode properties and supercapacitor performance is fundamental.

A large surface area combined with small diffusion paths of the nanostructured anodic oxides can favor fast redox reactions [31]. Besides, the excellent adherence to the substrate is an advantage in pseudocapacitive electrode architecture. When the metal oxide nanoparticles are used in the electrode, the weak adhesion between the nanoparticles and the current collector requires a binder agent. Otherwise, the charge/discharge processes can detach the layers decreasing the device’s performance [31]. The nanostructured oxide produced via anodization allows the fabrication of binder-free electrodes [31,125,126], reducing the synthesis steps. Additionally, the anodized metal substrate can act directly as the current collector [126].

Currently, the use of anodic oxides supercapacitor applications is in the development stage. Due to its high surface area with vertically aligned nanotubes, its chemical and thermal stability, and its wide potential window, the potential of TiO_2_-NT and its composites as electrodes for supercapacitors have been extensively investigated in latest years [31,48,120,125,126,127,128,129,130,131]. Copper hydroxides/oxides are also receiving attention due to their high theoretical specific capacitance, besides the advantages of the low-cost and abundance of copper [78,124,126,132]. Other studies reported the pseudocapacitive properties of anodic oxides produced from Ni [41,133], W [134], Nb [135], Mo [136]. All these papers described significant results and demonstrated the feasibility of these materials for energy storage applications. In general, these initial studies focused on optimizing the electrode architecture design and evaluating its electrochemical performance, capacitive and mechanical properties for a potential application in a supercapacitor. Despite enormous progress, there is still a lack of information concerning practical tests in supercapacitor devices.

The low number of studies reporting the tests of the anodic oxides as pseudocapacitors in supercapacitor devices were based on TiO_2_-NT [48], AAO [50], Cu(OH)_2_ [78], and NiO [41]. The first-two cases reported the use of TiO_2_-NT and AAO as supporting material for MnO_2_ in symmetrical supercapacitors. The other two applied Cu(OH)_2_ and NiO as the positive electrode in asymmetrical supercapacitors. Table 5 depicts the performance of these pseudocapacitive electrodes tested in symmetrical and asymmetrical supercapacitors (ASC) in water-based electrolytes in terms of their specific capacitance, energy density, and long-term stability. Note that the units are not standardized, so a direct comparison is not recommended.

By combining anodizing, heat treatment under ammonia atmosphere, and electrodeposition, Zhang et al. [48] prepared TiN_x_O_y_ electrodes loaded with MnO_2_ deposits and build a flexible symmetric superconductor to evaluate the performance of the material. The TiO_2_-NT arrays, fabricated by Ti anodization, was nitrated via annealed under ammonia flow at 800 °C for 3 h. Hence, MnO_2_ was electrodeposited from MnSO_4_ solution to form MnO_2_-TiN_x_O_y_ nanoarrays. An impressive energy and power densities were observed, and good cyclic stability (Table 5). According to the authors, TiNxOy nanoarrays’ conductivity enhanced significantly after nitridation compared with TiO_2_-NT, promoting a fast ion-exchange. Besides, the load with MnO_2_ improved the redox capacitance of the material.

The eco-friendly MnO_2_ is extensively applied as electrode material in supercapacitors due to its cost-effectiveness despite its low stability [120]. Besides TiO_2_-NT, AAO was employed as a substrate to MnO_2_ deposition. Gao et al. [50], for instance, build a supercapacitor device using a 3-D electrode consisting of MnO_2_-electrodeposited in AAO over FTO support. This electrode architecture was chosen to provide structural stability and high surface area. As a proof-of-concept, the authors built a symmetric self-membrane pseudocapacitor device by stacking two electrodes. They observed a superior performance of this material compared with other 3-D conventional electrodes and planar electrodes prepared via different methodologies [50].

To evaluate the use of nanostructured Cu(OH)_2_ as a flexible electrode in a supercapacitor device, Chen et al. [78] anodized Cu foil in alkaline media. So, the authors fabricated an asymmetric supercapacitor applying the Cu(OH)_2_/Cu as the positive electrode and activated carbon (AC) as the negative electrode, separated by polyvinyl alcohol/KOH gel as the electrolyte. This flexible and foldable homemade device delivered a high capacitance, superior rate capability, and cyclability (see Table 5) compared with other pseudocapacitive materials reported in the literature. Figure 5 illustrates the flexibility of the asymmetric supercapacitor device and voltammetric curves under different folding states. The remarkable results showed great potential for cost-effective commercial devices with non-toxic materials.

By combing anodizing and heat treatment, Cheng et al. [41] prepared NiO nanopetals on Ni substrate. They tested it as the positive electrode in asymmetric supercapacitor consisting of active carbon electrode as a negative electrode separated by a cellulose paper and KOH solution as the electrolyte. Although it exhibited excellent cycling stability, the specific energy density and power density were lower than other symmetric and asymmetric supercapacitors reported in the literature based on Ru, RuO, MnO_2_, and Fe_3_O_4_ electrodes. According to the authors, despite the lower performance, they found promising results considering the advantage of the low-cost, environmental-friendly, and simple fabrication of the NiO/Ni electrode.

### 6.2. Rechargeable Batteries

Batteries are devices that convert energy from a spontaneous electrochemical reaction into electrical power. They consist of one or more cells containing the anode, electrolyte, and cathode, and they are classified according to their materials and redox reactions. Rechargeable batteries are considered a sustainable form of store energy. Despite the lower power density and inferior long-cyclic performance than supercapacitors, they present a higher energy density, low self-discharge, and both high gravimetric and volumetric densities, mainly the lithium-ion batteries (LIBs) [32], currently the preferred battery-type for small electronic devices.

The most explored anodic oxide in batteries devices is recently based on TiO_2_ applied as anodes in ion batteries to overcome the technical issues during lithiation/delithiation cycling [137,138,139]. The major reports are devoted to LIBs [27,32,56,140,141] and sodium-ion batteries (SIBs) [28,142,143]. These studies reported the use of nanotubes or nanoporous TiO_2_ in pristine form, or doped (with Al, V, Nb, N, S), or in composites (with TiN, SnO_2_, CNT, CuO, CoO, Co_3_O_4_). Other nanostructured anodic oxides are also described in the literature, such as those obtained from anodization of Cu, Sn, W, Zn, and Nb foils in LIBs, SIBs, and Zn-ion batteries (ZIBs), as can be seen in Table 6, which shows the anodic oxides employed in batteries and their function. In general, these studies focused on improving battery performance by tuning the electrode properties, and evaluating their effects on storage capacity, rate capability, and long-term stability of the battery.

The search for alternative materials to replace conventional electrodes in LIBs and SIBs aimed to improve kinetic processes and storage capacity [59,144] and extend the device’s life cycle [145,146]. With a theoretical capacity of 330 mAh·g^−1^ and a good cycle life [140], the use of nanostructured anodic TiO_2_ in LIBs is also stimulated by the low volume change (≈4%) during the reversible intercalation of lithium ions into its structure [56,83,140,144]. The Li^+^ ions insertion from the cathode into the TiO_2_ anode proceed as follows [140]:TiO_2_ + xLi^+^ + xe^−^ → Li_x_TiO_2_(4)

Besides, especially in the anatase phase, TiO_2_ is considered the more electroactive host of Li insertion, with a theoretical electromotive force of ≈1.7 V (vs. Li/Li^+^) [27,83,140,145]. This low voltage required for lithium insertion improves the battery safety [56], and prevents the formation of an unstable solid electrolyte interface (dendrites) on the electrode surface, reducing the risk of a short circuit during overload [144]. The TiO_2_-NT also promotes fast kinetics and improves capacity retention [57].

Sugiawati et al. [140], for instance, used anodized TiO_2_-NT to build an all-solid-state LIB consisting of a TiO_2_-NT anode covered with a thin layer of polymer electrolyte (MMA-PEG) as electrolyte and separator; and a layer of LiFePO_4_ as a cathode. This system obtained high-capacity retention of 97.35% and Coulombic efficiency of 96.78% after 50 cycles. In another study carried out by Zhang et al. [56], the self-doping of TiO_2_-NT with oxygen vacancies or Ti^3+^ ions fabricated by cathodic pulsation during the anodization made it possible to obtain a battery with a high specific initial discharge capacity of 1355 mA·h·cm^−2^. This value was remarkably larger than 338 mA·h·cm^−2^ obtained using the bare anodized TiO_2_ anode. The specific capacity of the electrodes was maintained at 623 mA·h·cm^−2^ after 100 cycles. These gains were attributed to the excellent retention of the tubular morphology and the secondary growth of the self-doped TiO_2_-NT arrays.

Other strategies used to enhance discharge capacity, cycling behavior, rate capacity, and electronic conductivity of LIBs based on TiO_2_-NT were: (i) the annealing of Ti sheets before anodizing to remove residual stress among the Ti atoms and facilitate the synthesis of the nanoporous TiO_2_ anodes [27]; (ii) the use of alloys such as the Ti-Nb alloy in the anodic synthesis of TiO_2_-NT layers, where the substitution of Ti^4+^ by Nb^5+^ cations and the enrichment of Nb on the top surface of the nanotubes increased electronic conductivity [57]; (iii) surface modification with other oxides with good gravimetric capacities, such as Co_3_O_4_ and CuO deposited by spray-coating technique [83]; (iv) the fabrication of composite materials based on TiO_2_-NT, as illustrated in Figure 6, which schematizes the synthesis of CNT@TiO_2_/CoO NT anode for LIBs, as proposed by Madian et al. [59].

As an alternative to LIBs, SIBs are receiving much attention in modern studies, since these batteries are based on sustainable precursors and more secure raw material supplies [151,152]. However, despite the similar chemistry of lithium and sodium, the large radius of the sodium ion compromises its reversible intercalation into the conventional graphite anodes. Therefore, the controlled morphology of TiO_2_-NT is an advantage for use as the anode in SIBs, providing large size channels with suitable interstitial sites for the accommodation of guest ions and diffusion paths for sodium ions. In addition, the TiO_2_ exhibits a low insertion potential (−0.7 V vs. Na/Na^+^) and a large theoretical capacity (335 mAh·g^−1^) [28].

A drawback of SIB technology is the slow kinetics of the Na^+^ electrode [142]. In this sense, some of the strategies to enhance this are the doping of TiO_2_ [25] or lithiated SIB use [142]. Ni et al. [28] doped TiO_2_-NT with S during the annealing at 500 °C under Ar gas containing S vapor to improve the SIB system’s storage properties. The authors observed the highest sodium storage in terms of high capacity (320 mAh·g^−1^), an ultra-stable cycle (91% retention in 4400 cycles), and adequate rate capacity (167 mA·h·g^−1^ to 3.35 A·g^−1^). The anodization and sulfurization procedure to prepared S-doped TiO_2_-NT improved the battery’s electrochemical capacity and sodium storage. The results were attributed to the nanotubes’ arrangement, the doping effect on electronic properties, and the increased kinetic stability. Cha et al. [142] showed that the Li^+^ pre-insertion onto TiO_2_-NT anodes in SIBs produced a high reversibility capacity, cycle stability, and a high-rate performance. According to the authors, this approach expanded the anatase lattice onto Li-TiO_2_ nanotubes facilitating Na^+^ cyclability and increasing conductivity [142].

Regarding other anodized oxides, some studies described the use of nanostructured WO_3_ and Sn@Cu_x_O in LIBs [72,79], SnO_x_, and Nb_2_O_5_ in SIBs [148,149,151], and ZnO in ZIBs [76]. Due to its high theoretical capacity (696 mA·h·g^−1^) [72], WO_3_ is also a potential material for anodes in LIBs. The drawback of its application is the aggregation of the WO_3_ nanomaterials, which reduces its capacity. To minimize this effect, Yang et al. [72] produced a 3D network of WO_3_ nanosheet arrangements via two-step anodization of W foil and decorated it with Ag-NPs to enhance electronic conductivity. The Ag-NP was loaded into the WO_3_ layer via sol-gel and spin-coating techniques. The resulting Ag-decorated WO_3_ electrode maintained a rate performance of 681 mA·h·g^−1^ after 150 cycles and a discharge capacity of 917 and 370 mA·h·g^−1^ at current densities of 0.1 and 1.0 mA·cm^−1^, respectively.

For use as the anode in LIBs, Kim et al. [79] fabricated a heterostructured copper oxide nanowires via Cu anodization and electrodeposited Sn over it from a Sn_2_P_2_O_7_ precursor. According to the authors, the eco-friendly Sn anodes exhibits higher theoretical specific capacities (994 mA·h·g^−1^) but expand up to 300% of their initial volume during Li^+^ ion intercalation. On the other hand, copper oxides exhibit a moderate theoretical specific capacity (675 mA·h·g^−1^) and a lower tendency to change their volume drastically. Hence, the authors expected that an Sn/Cu_x_O nanowire anode enhanced the charge processes’ efficiency in LIB applications. The results revealed that the composite lead to a reversible discharge capacity of 772.5 mA·h·g^−1^ and stable cycle retention for 100 cycles. The authors concluded that Cu_x_O nanowires in the composite increased the discharge capacity providing direct electronic pathways, alleviated the stress applied to the electrode, and improved the material’s stability even after multiple cycles.

Other synthetic routes involving anodic oxides such as SnO_2_ [148] and Nb_2_O_5_ [149] are being developed to produce nanostructured films with better storage capability for Na^+^ ions in SIBs. Both materials were applied as the anode in SIB systems exhibiting high-rate capacity and cycle stability [148,149]. Bian et al. [148] used a water immersion approach to crystallize anodized SnO_2_ at low-temperature and produce a mesoporous rutile-SnO_2_. Besides developing a crystallization methodology at low temperatures, the system delivered a superior electrochemical performance for Na^+^ ion storage (514 mA·h·g^−1^ after 100 cycles at 0.1 C). The promising results were ascribed to the high surface area with a larger number of reactive sites, short diffusion length for Na^+^ ions transport, and electrical conductivity improved by the crystallization.

After previous studies had demonstrated that Nb_2_O_5_ exhibited a high storage capacity in LIBs, Ni et al. [149] tested this semiconductor as an intercalation anode for SIBs. For this, they anodized Nb foil and enriched the amorphous nanoporous Nb_2_O_5_ array with H by heat treatment under Ar/H_2_ atmosphere at 450 °C and 600 °C. The comparison between amorphous (400 °C) and the crystalline (600 °C) samples showed that the amorphous material leads to outstanding performance for SIBs, with a high reversible capacity of 185 mA·h·g^−1^ at 0.5 C in the potential range of 0.1–2.5 V (vs. Na^+^/Na) and a stable capacity of 109 mA·h·g^−1^ at 5 C upon 3000 cycles. According to the authors, the amorphous material offered higher activity and sustainability than the crystalline one. Furthermore, the carrier transport could be enhanced with the addition of non-metals elements, which introduced oxygen vacancies and increased conductivity.

Zn-ion batteries are considered a green energy system [76] using less-flammable electrolytes and low-toxic materials. Kim et al. [76] used the periodic anodizing technique to produce ultra-stable Zn anode in a hexagonal pyramid arrangement with a functionalized ZnO layer for a ZIB device. This structure inhibited the growth of Zn dendrites, which damage the electrode stability. The Zn@ZnO/MnO_2_ battery exhibited outstanding long-term cyclability and a coulombic efficiency ≈99% after 1000 cycles at a current density of 9 A·g^−1^. The remarkable results were attributed to the periodic process of synthesis that optimized the Zn@ZnO interface maximizing the electroactive surface area and improved the corrosion resistance.

Besides anode materials, some studies also described other anodic oxides’ functionalities, such as cathode [147] and solid-state separator [150]. Xia et al. [147], for instance, fabricated oxygen-deficient nanoporous Ta_2_O_5_ film by anodization of Ta foil aiming to obtain flexible and durable 3D electrodes for LIBs. The Li^+^ ion storage into Ta_2_O_5_ proceed according to [147]:Ta_2_O_5_ + x(Li^+^ + e^−^) ↔ Li_x_Ta_2_O_5_(5)

The theoretical capacity of 482 mA·h·g^−1^ for Ta_2_O_5_ exceeds the graphite’s capacity, showing this semiconductor’s potential for batteries. By producing a coin cell using Ta_2_O_5_ film as the cathode, Li metal as an anode, and LiClO_4_ solution in propylene carbonate as the electrolyte, the authors observed a lithium capacity similar to the theoretical value and considerable cycling stability over 8000 cycles at 5 C rate. This performance was ascribed to the amorphous structure with oxygen deficiency and nanoporous architecture, which improved the electrode’s transport kinetic and structural integrity.

The lithium–sulfur (Li–S) battery is an up-and-coming technology for high-energy storage systems. However, it presents limitations regarding the safety related to lithium metal anode, the low conductivity of sulfur, and the formation of the lithium polysulfides soluble in the electrolyte, which decreases the performance of the battery [150]. Due to rigidity and non-flammability, Wang et al. [150] tested commercial AAO membranes as porous solid-state separators in Li–S batteries. The results showed a cell stable performance of 480 cycles with a capacity retention of 50.4% at 2 C rate, better than using Celgard^®^ separator, the conventional separator for LIBs. The AAO membrane offered fast pathways for lithium-ion transport and improved battery safety.

The use of AAO as a template to drive the synthesis of the other materials was also reported in LIBs [153,154,155], SIBs [156], and Li–S batteries [157]. Yoo et al. [153] synthesized AAO membranes to prepare SiO_2_ hollow nanorods for LIB cathode. Li et al. [154] applied AAO to produced 3D nanostructures of LiAlO_2_-modified LiMnPO_4_ for LIB cathode. Fang et al. [155] used AAO to fabricate Sn nanowires encapsulated in Al_2_O_3_ tubes in a carbon matrix for LIB applications. In a SIB system, Xu et al. [156] synthesized AAO for electrodeposited Ni nanopillar arrays. After removing AAO, TiO_2_ was deposited over Ni arrays by Atomic Layer Deposition (ALD) to fabricate Ni-TiO_2_ nanoarrays cathodes. Hu et al. [157] utilized commercial AAO as a template to prepare a carbon composite material with sulfur for a Li–S battery application. In this case, the carbon nanotubes were deposited on the pore-walls of the AAO template by chemical vapor deposition (CVD) from a mixture of C_2_H_2_ and N_2_. The AAO/CNT membrane was dipped into a molten S-1,3-diisopropenybenzene (S-DIB) to fill the channels. After the removal of AAO, the S-DIB@CNT composite was applied as the cathode in the Li–S battery. In some of those studies, AAO templates were applied as a viable strategy to tailor a specific nanoarchitecture design of the battery’s electrode. For other studies in which AAO was an active part of the electrode structure, diffusional and electronic transport were enhanced during the battery operation, i.e., during the charge–discharge cycle [154].

Nanostructured anodic devices present several advantages for rechargeable batteries application. The strategy is to use nanostructure-designed anodic oxide, applying in most of the cases top-down manufacturing to build electrodes can enhance ions’ transport kinetics during the battery operation [155], providing an orientated transport improvement through the electrodes morphology/architecture [154]. Otherwise, as discussed in [158,159], if the nanoscale electrode materials match with the wavelength of electrons, phonons, and other material constituents, quantum effects might occur, improving the electrical conductivity. Some examples of these approaches may be found in [153,154,156,160,161,162,163,164,165,166,167,168]. Although the above articles point to exciting strategies for improving battery conductivity, these technologies are still in fundamental science and will not be discussed in depth in this applied material review.

In summary, this section presents the advantages of anodic oxide nanostructures with a large specific surface area providing faster electron transport and high ion accessibility maximizing the power and energy densities. Moreover, the nanostructured morphology provided by these hierarchical structures compared to flat electrodes could play a role in avoiding electrode mechanical failure due to volume changes during the charging and discharging process.

All these adopted strategies in developing both batteries and supercapacitor devices revealed remarkable results concerning nanostructured anodic oxides. They can contribute to the development of green energy-storage systems with high efficiency, long-term stability, and safety. As a consequence, they can provide significant improvements for the industrial and commercial sectors.

## 7. General Remarks

Besides the advantages of the fabrication, such as scalability, easy synthetic route, and control of properties, the results demonstrated that the nanostructured anodic oxides presented additional benefits that can lead to improvements in the fabrication of devices for energy conversion and storage. In energy-conversion devices that operate via photo- and photoelectrochemical processes, such as the solar cells used for electricity production and the PEC water-splitting devices used for H_2_ generation, the nanostructured oxides in the electrodes provide significant contributions to increasing efficiency and performance, which can make these devices cost-effective.

In these cases, the electrodes’ properties of interest were tuned to increase each device’s performance by improving the photoconversion efficiency and resistance to photo- and electrochemical corrosion. The TiO_2_-NT and nanoporous WO_3_ were the most explored anodic oxides in the energy-conversion applications under different architectures. In DSSC and PEC devices, TiO_2_-NT anodes stood out, although WO_3_ electrodes also demonstrated a good performance in PEC devices. ZnO and Cu_x_O nanostructures also exhibited promising results in PEC applications. The studies showed that nanotubes’ unidirectional orientation and nanoporous structures enhanced the charge transfer and mass transport processes. The large specific surface area also favored the diffusion and adsorption/desorption of species. Evidence indicated that heterostructures combining NPs and NTs facilitated charge transfer, diminishing the recombination rate. Templates and scaffolds of AAO also provided significant improvements in the synthetic route of other materials.

Regarding the energy conversion devices operating via electrochemical processes, fuel cell technology is still developing. However, nanostructured anodic oxide films’ utilization as supporting material for the catalyst, such as Pt, Pd, or NiO, proved to influence the activity and stability of the electrodes in corrosive media or high temperatures, thereby improving the overall performances of the devices. The larger surface area provided a high number of active sites for the catalytic processes, thereby increasing efficiency. The chemical stability and resistance to corrosion of the anodized oxides increased the long-term stability of the catalyst. TiO_2_-NT was tested in PEMFC, DMFC, DFAFC, and MFC. The biocompatibility of TiO_2_ makes this oxide the active catalyst in the MFC. Other nanostructures based on NiO, ZrO_2_, and AAO were also reported, but with different functionalities—for instance, ZrO_2_-NT as a solid electrolyte in a SOFC device.

In energy-storage devices such as supercapacitors and batteries, the anodic oxide significantly contributed to the devices’ electrode architectures and performances. In supercapacitors, the growth of the oxide directly from the metal substrate, which can act as the current collector, avoids the use of a binder agent between the layers. This action is an advantage via reducing the time of the synthesis. Additionally, the high surface area increases the capacity of the material to store charge on its surface. Regarding the practical applications, the pseudocapacitive properties of Cu(OH)_2_ and NiO nanopetals were evaluated in an asymmetric supercapacitor with promising results. In symmetric supercapacitor, TiO_2_-NT and AAO were employed as support for MnO_2_, enhancing the electrode’s stability and performance.

In rechargeable batteries, the nanostructured architectures grown on substrates have gained attention as binder-free electrodes, with large surface areas and open frameworks allowing for efficient mass transport and ion diffusion through the nanochannels. With low expansion during the ions’ intercalation, the use of TiO_2_-NT stood out in Li^+^ and Na^+^ ion batteries. Still, considerable attention was also given to other anodic oxides, such as Cu_x_O, WO_3_, SnO, Nb_2_O_5_, and Nb_2_O_5_. The properties of these materials were tuned to improve the batteries’ efficiency, cyclability, and safety.

## 8. Technological Aspects and Future Perspectives

This review focused on presenting works involving nanostructured anodic oxides manufactured by anodizing in practical applications of energy devices. Since the synthesis of this type of material has been known for decades, is well-established, and is recognized in a series of applications in different areas, we opted to bring a compilation of more recent works while focusing on the applications instead of the synthesis. This review’s main goal was to assess this class of materials’ practical contributions to technological applications, regarding the worldwide interest in scientific research to break academic barriers and make usable devices and technologies that contribute to society’s development.

In 2015, the United Nations General Assembly established the General Objectives for Sustainable Development. Among them, we single out Objective 7 (Industry, Innovation, and Infrastructure) and Objective 9 (Accessible and Clean Energy). Objective 7 is strengthening international cooperation until 2030 by allowing access to research and seeking development and investment in clean energy technologies. In this sense, PV and PEC devices’ development is important due to using the most abundant energy source on the planet, the sun, and its use in hydrogen production and consequent conversion into energy. On the other hand, one of the goals of Objective 9 of Agenda 2030 is to increase efficiency in using natural resources and promote environmentally friendly and clean industrial technologies and processes. From this perspective, the anodic oxides topic is so rich that its classification according to the scale used to assess the maturity level of a particular technology, the technological readiness level (TRL) [169], goes from basic research (1) to the highest level (9)—where the aim is mass production and the commercial use of these devices.

From a technical point-of a view, the big challenge of the materials in the energy field is related to the synthesis of new materials with specific properties and different components to create completely functional devices. For instance, the fundamental problems regarding photoelectrodes’ stability and the high photoconversion rates related to the domain of defect carriers’ species present in the devices’ electrodes and membrane components can be classified as TRL-1 and TRL-2. On the other hand, TRL-3 and TRL-4 are associated with the adequate mounting of the different device parts, maintaining the specific properties of each piece working as they are designed, moving close to a proof-of-concept device. Binding the different parts of the device without decreasing the performance is a great challenge. TRL-5 to TRL-9 are related to the production chain and commercial use of the final device. In the authors’ opinion, this review presents the related literature regarding TRL-2 and TRL-3 used for energy converting and storage devices applying (directly or indirectly) some anodic oxide in their conception.

The oxide synthesis process involves clean, easy, and fast methods with high surface area production, high chemical stability, and increased adhesion to the substrate. Despite these advantages, at an industrial scale, the preference is still for other types of synthesis involving chemical, physical, or hybrid methods instead of the electrochemical method. As observed in previous sections, the electrochemical synthesis also exhibits excellent potential for improving and controlling the device’s properties with gains in efficiency and durability. When combined with other techniques, it can offer improved device performance and reduce costs.

For most applications, the use of nanostructured anodic oxides is in the development stage and is still far away from commercial applications, despite the numerous advantages listed here. Looking back, it is worth remembering that for many years, energy from fossil fuels was the only alternative. After countless studies and the presentation of several possibilities, it is up to the industrial sector to choose among the options—those that result in optimal cost–benefit ratios. Regarding the application of the anodic oxides, their remarkable characteristics and their potential for use in energy production and storage devices are some of the possibilities that materials engineering offers for the future.

## Data Availability

Data sharing not applicable.

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
