# Peer review of "The Use of Anodic Oxides in Practical and Sustainable Devices for Energy Conversion and Storage"

_materials, 2021, doi:10.3390/ma14020383_

Round 1

Reviewer 1 Report

In this paper, Santos et al., reviewed the application of the anodic oxides in improving the efficiency and performance of the devices for energy conversion and storage applications. In particular the review covers i) general aspects of anodic oxide synthesis, ii) photovoltaic devices for energy conversion, iii) photoelectrochemical devices for H2 production, iv) electrochemical devices for energy conversion and v) energy storage devices.

The review offers an extensive overview of using anodic oxides, it is rich in details and examples.

1) The main issue, in my view, is the lack of discussion. If possible, before the section dedicate to “conclusion”, I would recommend to add a section where the authors present their point of view and drawing connections.

2) Also, if possible, I would like to have a list of the acronyms used throughout the manuscript. Sometime it is challenging keep all of them in mind during the reading.

Few additional comments.

3) In the figure 2

The schematic of Ti wires and glass tubes in the panel E and F incudes some labelling printed too small.

4) In Figure 4

In the power curve, the authors might have indicate “mV” instead of “mW”

5) Line 793

The authors have written: “MFCs are devices producing electricity from organic wastes”.

MFCs can produce electricity from organic substrates, not just waste. Please change this accordingly.

6)Figure 6

Some elements in this figure are printed too small

Author Response

Answer to Reviewer 1    

Dear reviewer,

We thank you immensely for the contributions made to our manuscript. We have tried to answer all the questions raised, and we modified the manuscript accordingly. All information changed and added was highlighted in yellow. We hope that this new version of the manuscript fits your expectations and agrees with your notes, being now suitable for publication in Materials Journal.

The answers were listed below:

Question # 1- The main issue, in my view, is the lack of discussion. If possible, before the section dedicate to "conclusion", I would recommend to add a section where the authors present their point of view and drawing connections

Answer: We appreciate the above suggestions. We emphasize some suggestions during the text (highlighted in yellow). However, to not compromise the finished text structure and author's style, discussion content and connections were added to the manuscript file as a new section ("Technological Aspects and Future Perspective").

Question # 2 - Also, if possible, I would like to have a list of the acronyms used throughout the manuscript. Sometimes it is challenging to keep all of them in mind during the reading.

Answer: An Abbreviation list was added.

Question # 3 - In the figure 2

The schematic of Ti wires and glass tubes in the panel E and F includes some labelling printed too small.

Answer: The requested changes were made in Figure 2.

Question # 4 - In Figure 4

In the power curve, the authors might have indicate "mV" instead of "mW"

Answer: The reviewer is right. There was probably a mistake in using the power unit. We corrected and left the change made indicated in the figure caption.

Question # 5 - Line 793

The authors have written: "MFCs are devices producing electricity from organic wastes". MFCs can produce electricity from organic substrates, not just waste. Please change this accordingly.

Answer: The word "substrate" is used in a different sense in the text than the reviewer proposes, attributed to the solid starting materials used in the electrodes' synthesis. To avoid confusion, we replaced the term "organic waste" in the text with "organic compounds."

Question # 6 - Figure 6: Some elements in this figure are printed too small

Answer: The requested changes were made in Figure 6.

Reviewer 2 Report

The work entitled "The use of anodic oxides in practical and sustainable devices for energy conversion and storage" reported an overview on the importance and usage of nanostructured anodic oxide films to overcome the limitations of the energy conversion and storage devices. In general, this review provides a well-designed study covering the main aspects of the design and application of the anodic metal oxide in the energy storage devices.

1- However, it is necessary to clarify the novelty and the main goal of the review in the abstract part and as well as the claim at the end of introduction.

Moreover, there are some points must be highlighted and addressed as follows:

2- “Abstract” part:

The abstract needs more information and clarification related to the main goal, novelty of the work, and future.

3-“Introduction” part:

This part needs to carefully revise.

The author start this part directly with his goal "claim of the work, as he mentioned "In this review, we will discuss.....". Thus it is recommended that:

  • the author should start with this section with general information that reflect the main problem and challenges, available solution, and the importance of materials and techniques.
  • review the development of the current research problem, challenge and current solutions
  • Advantage and disadvantage leading to the current study, this means the authors highlight the development of their research and gave the challenge and solution
  • More attention should pay to the nanostructure material used in this field of energy application
  • It is recommended to add photos in the first scheme show the application if possible.
  • There are some relevant and updated citations proposed to assist and guide the authors in this work:

Applied Materials Today 19, (2020) 100590; ACS Appl. Energy Mater. 2020, 3, 9, 9168–9181; National Science Review 2020. 7 (5), 863-880; Batteries & Supercaps 2020, 3 (1), 76-92; Nano-Micro Letters 2019, 11 (1), 84; Scientific reports 2018, 8 (1), 1-14; Chemical Engineering Journal 2017, 313, 83-98; Journal of Power Sources 2016, 330, 292–303

4- "2. General aspects of anodic oxide synthesis for energy applications":

  • It is recommended to add a general chart showing the various fabrication and adjustment steps.
  • Additionally, this section needs further explanation of one technique used.
  • The author has provided short paragraphs on each page, so it is recommended that you provide three or four paragraphs per page.

5- "Regarding "3. Photovoltaic devices for energy conversion: Solar cells",

  • the author should clarify the main challenges in this field and the needs for nanofabrication.
  • The author mentioned"3.2. Other functionalities of anodic oxides in Silicon, PSC, and OPV", where is the "3.1. ....."
  • The importance of materials should be cleared.

6- For all sections of application,

  • The author should highlight the main problem and challenge, and then the importance of nanomaterials to solve this problem.
  • The author has to present the effect of nanomaterial structure on the performance
  • The author has provided short paragraphs on each page, so it is recommended that you provide three or four paragraphs per page.

7- There are many applications with lack of information, it is recommended to focus in more detail on a specific one or two applications with highlighting others

8- Conclusions need more attention and review to clarify the importance of this review, the recommendation, main outcomes, and the future of implementation depending on the proposed methods

9- Language needs revision

Author Response

Answer to Reviewer 2

Dear reviewer,

We thank you immensely for the contributions made to our manuscript. We have tried to answer all the questions raised, and we modified the manuscript accordingly. All information changed and added was highlighted in yellow. We hope that this new version of the manuscript fits your expectations and agrees with your notes, being now suitable for publication in Materials Journal.

The answers were listed below:

Question # 1 - However, it is necessary to clarify the novelty and the main goal of the review in the abstract part and as well as the claim at the end of introduction.

Answer: We are grateful for the review and suggestions. We have made the modifications at the beginning of the abstract and at the end of the introduction highlighted in the text.

Question # 2 - “Abstract” part:

The abstract needs more information and clarification related to the main goal, novelty of the work, and future.

Answer: The suggested changes were made and highlighted in the text.

Question # 3 -“Introduction” part:

This part needs to carefully revise.

The author start this part directly with his goal "claim of the work, as he mentioned "In this review, we will discuss.....". Thus it is recommended that:

3.1 the author should start with this section with general information that reflect the main problem and challenges, available solution, and the importance of materials and techniques.

Answer: We are grateful for the reviewer's careful review regarding the introduction of this manuscript. As suggested, we begin the text with the presentation with general information. However, more profound changes involving the challenges, solutions, and importance of the materials were made throughout each section. All changes have been highlighted in the text.

3.2 Review the development of the current research problem, challenge and current solutions, advantage and disadvantage leading to the current study, this means the authors highlight the development of their research and gave the challenge and solution

Answer: After re-reading the text, we realized that some of the points mentioned by the reviewer were not properly available, as the comment left. Therefore, some modifications were made, which are highlighted throughout the text. Particularly about the advantages, these were added at the beginning of section 2.

3.3 More attention should pay to the nanostructure material used in this field of energy application

Answer: We thank you for the references you pointed out. After reading, we identified that perhaps this comment is associated with the different types of nanostructured materials used mainly in lithium-ion batteries' research. Despite the high amount of materials and their extraordinary characteristics, we emphasize that the focus was on the applied nanostructured oxides grown exclusively by electrochemical methods in this manuscript. Other excellent bibliographies show the significant advances in these materials for anodes and cathodes and are even at higher technology readiness levels (TRL) and application than the anodic oxides discussed in this work.

3.4 It is recommended to add photos in the first scheme show the application if possible.

There are some relevant and updated citations proposed to assist and guide the authors in this work:

Applied Materials Today 19, (2020) 100590; ACS Appl. Energy Mater. 2020, 3, 9, 9168–9181; LIB

National Science Review 2020. 7 (5), 863-880; Batteries & Supercaps 2020, 3 (1), 76-92; LIB

Nano-Micro Letters 2019, 11 (1), 84; Scientific reports 2018, 8 (1), 1-14; LIB

Chemical Engineering Journal 2017, 313, 83-98; Journal of Power Sources 2016, 330, 292–303

Answer: We are grateful for the reference indications to redesign scheme 1. As suggested, we added representative photos of the applications discussed in the text.

Question # 4 - General aspects of anodic oxide synthesis for energy applications": It is recommended to add a general chart showing the various fabrication and adjustment steps.

Additionally, this section needs further explanation of one technique used.

The author has provided short paragraphs on each page, so it is recommended that you provide three or four paragraphs per page.

Answer: The use of metal oxides for energy applications is vast and comprises distinct synthetic techniques from chemical, physical, and electrochemical routes. Also, the topic "anodic oxides" is no longer a novelty in materials science. There are excellent texts available on the different methodologies used to synthesize these materials, which we cite at the end of this letter. It is important to stress that this manuscript covers the papers describing anodic oxides produced exclusively via anodization technique from 2015 to 2020. Besides the anodization, we used other criteria to select the articles focused on the application instead of synthesis. Then, studies where the material was not tested in a practical or simulated energy device, were disregarded. It would not be fair to expand the description of the materials that might not be relevant for the review subject. Otherwise, we might be at risk of a restricted written space for a proper discussion of these papers. So we choose not to extend the review so much and prioritize the main subject chosen by us.

Question # 5 - "Regarding "3. Photovoltaic devices for energy conversion: Solar cells", the author should clarify the main challenges in this field and the needs for nanofabrication.

Answer: The importance of the materials, main challenges, and the needs for nanofabrication in solar cells are described in section 3 and 3.1, highlighted throughout the manuscript.

Question # 6 - The author mentioned "3.2. Other functionalities of anodic oxides in Silicon, PSC, and OPV", where is the "3.1. ....."

The importance of materials should be cleared.

Answer: The materials' importance is described in each section; the excerpts were highlighted throughout the manuscript.

Question # 7 - For all sections of application, the author should highlight the main problem and challenge, and then the importance of nanomaterials to solve this problem.

Answer: In all sections focused on applications, we highlight the text excerpts referring to the main problem, challenges, and the importance of nanomaterials for the solution of the problems addressed.

Question # 8 - The author has to present the effect of nanomaterial structure on the performance.

The author has provided short paragraphs on each page, so it is recommended that you provide three or four paragraphs per page.

Answer: We understand the question of the referee. However, we choose to write a review paper different from others. This review's main objective is to pinpoint the anodic oxide-based materials used to develop alternative energy devices and close to an applied device. It is not the purpose of this review to track all effects of the nanomaterial structure produced and check their performance. We rechecked the introductory part and made some adjustments to ensure that this message was there.

Question # 9 - There are many applications with lack of information, it is recommended to focus in more detail on a specific one or two applications with highlighting others

Answer: Several review articles in literature [1-13] focus on one or a few applications of anodic oxides and describe such applications. Such reviews are also referred to below, but that was not our proposition for this particular review article. Some of them were inserted into the review reference table.

References:

[1] GIZIŃSKI, Damian et al. Nanostructured Anodic Copper Oxides as Catalysts in Electrochemical and Photoelectrochemical Reactions. Catalysts, v. 10, n. 11, p. 1338, 2020. https://doi.org/10.3390/catal10111338

[2] SANTOS, Abel; KUMERIA, Tushar; LOSIC, Dusan. Nanoporous anodic aluminum oxide for chemical sensing and biosensors. TrAC Trends in Analytical Chemistry, v. 44, p. 25-38, 2013. https://doi.org/10.1016/j.trac.2012.11.007

[3] KUMERIA, Tushar; SANTOS, Abel; LOSIC, Dusan. Nanoporous anodic alumina platforms: engineered surface chemistry and structure for optical sensing applications. Sensors, v. 14, n. 7, p. 11878-11918, 2014. https://doi.org/10.3390/s140711878

[4] SANTOS, Abel; KUMERIA, Tushar; LOSIC, Dusan. Nanoporous anodic alumina: a versatile platform for optical biosensors. Materials, v. 7, n. 6, p. 4297-4320, 2014. https://doi.org/10.3390/ma7064297

[5] LOSIC, Dusan et al. Titania nanotube arrays for local drug delivery: recent advances and perspectives. Expert opinion on drug delivery, v. 12, n. 1, p. 103-127, 2015. https://doi.org/10.1517/17425247.2014.945418

[6] SANTOS, Abel et al. Nanoporous anodic alumina barcodes: toward smart optical biosensors. Advanced Materials, v. 24, n. 8, p. 1050-1054, 2012. https://doi.org/10.1002/adma.201104490

[7] MEBED, AbdElazim M. et al. Review on the Formation of Anodic Metal Oxides and their Sensing Applications. Current Nanoscience, v. 15, n. 1, p. 6-26, 2019. https://doi.org/10.2174/1573413714666180817130835

[8] LI, Huan-Huan et al. Titanium oxide nanotubes prepared by anodic oxidation and their application in solar cells. Acta Physico-Chimica Sinica, v. 27, n. 5, p. 1017-1025, 2011.

[9] WU, Hui et al. Ordered organic nanostructures fabricated from anodic alumina oxide templates for organic bulk‐heterojunction photovoltaics. Macromolecular Chemistry and Physics, v. 215, n. 7, p. 584-596, 2014.https://doi.org/10.1002/macp.201300766

[10] AWAD, Nasser K.; EDWARDS, Sharon L.; MORSI, Yosry S. A review of TiO2 NTs on Ti metal: Electrochemical synthesis, functionalization and potential use as bone implants. Materials Science and Engineering: C, v. 76, p. 1401-1412, 2017.

[11] PARAMASIVAM, Indhumati et al. A review of photocatalysis using self‐organized TiO2 nanotubes and other ordered oxide nanostructures. small, v. 8, n. 20, p. 3073-3103, 2012.

[12] Trivinho-Strixino, F.; Santos, J.S.; Souza Sikora, M. Electrochemical Synthesis of Nanostructured Materials. In Nanostructures; 2017; pp. 53–103 ISBN 9780323497831.

[13] CHEN, Y. ZHI et al. Anodized metal oxide nanostructures for photoelectrochemical water splitting. International Journal of Minerals, Metallurgy and Materials, v. 27, n. 5, p. 584–601, 2020. https://doi.org/10.1007/s12613-020-1983-6

Question # 10 - Conclusions need more attention and review to clarify the importance of this review, the recommendation, main outcomes, and the future of implementation depending on the proposed methods.

Answer: As suggested, we modified the text's structure by renaming the Conclusion to General Remarks. The recommendation, primary outcomes, and the future of implementation were discussed in the new section entitled "Technological Aspects and Future Perspectives."

Question # 11 - Language needs revision

Answer: We thank you for the revision pointed out. A language adjustment of the text was carefully performed.

Reviewer 3 Report

Present article is rather good written and might be interesting for the scientists working in different spheres of chemistry and materials science. 

The data described are structured well and clear. In my opinion in the chapter 6.1 Supercapacitors authord need to stress that everything described is actual for the water-based electrolyte. Authors also need to mention that carbon nanostructured (e.g. nanotubes or graphene) are frequently used together with oxides and by themselves can provide supercaspacitance [https://doi.org/10.1179/1753555714Y.0000000187].

In the case of metal ion batteries it would be good to mention the general routes to increase the conductivity of the systens by mixing with carbon forms.

Unfortunaly, potassium ion batteries are absent in the paper. So, it would be good the layered oxides using there. The last paper on it is  [https://doi.org/10.1016/j.ensm.2020.09.010].

Author Response

Answer to Reviewer 3

Dear reviewer,

We thank you immensely for the contributions made to our manuscript. We have tried to answer all the questions raised, and we modified the manuscript accordingly. All information changed and added was highlighted in yellow. We hope that this new version of the manuscript fits your expectations and agrees with your notes, being now suitable for publication in Materials Journal.

The answers were listed below:

Question # 1 - The data described are structured well and clear. In my opinion in the chapter 6.1 Supercapacitors authors need to stress that everything described is actual for the water-based electrolyte.

Answer: This information was added to the manuscript.

Question # 2 - Authors also need to mention that carbon nanostructured (e.g. nanotubes or graphene) are frequently used together with oxides and by themselves can provide supercapacitance [https://doi.org/10.1179/1753555714Y.0000000187].

Answer: This information and the reference were added to the manuscript.

Question # 3 - In the case of metal ion batteries, it would be good to mention the general routes to increase the conductivity of the systems by mixing with carbon forms.

Answer: This information was added to the manuscript and is highlighted in Yellow color.

Question # 4 - Unfortunaly, potassium ion batteries are absent in the paper. So, it would be good the layered oxides using there. The last paper on it is [https://doi.org/10.1016/j.ensm.2020.09.010].

Answer: The K-ion batteries are an essential class of batteries due to their low-cost and abundant potassium sources. However, we did not find recent papers describing the use of anodic oxide produced by anodization in such application. It is the reason why this type of battery was not mentioned in the manuscript since this review is dedicated to the nanostructured anodic oxides fabricated via anodic oxidation of metal substrates. We also checked the references cited in this review paper (Liu2021) that you mentioned. We didn’t find papers with anodized oxides either. The oxides cited there were synthesized by different methodologies but none using an anodization procedure.

Reviewer 4 Report

In this article, the authors present a review of nanostructured anodic oxides as electrode material for energy conversion and storage devices. The authors emphasized the electrochemical properties of promising oxides via photo, photoelectrochemical and electrochemical processes. Overall, this manuscript is a fair review on metal oxides. The reviewer has the following concerns that need the authors to clarify before this article can be considered to be published in Materials.

  1. Certain state-of-art materials as anodes in LIB are missing, for example, Dual-transition-metal oxides:

The authors need to add more references such as:

doi.org/10.1016/j.jallcom.2016.12.094
doi.org/10.1016/j.jallcom.2019.151691

  1. Citations regarding AAO used in Li-S batties are also missing, such as AAO as a template. Authors need to add those references and discussions.

Author Response

Answer to Reviewer 4

Dear reviewer,

We thank you immensely for the contributions made to our manuscript. We have tried to answer all the questions raised, and we modified the manuscript accordingly. All information changed and added was highlighted in yellow. We hope that this new version of the manuscript fits your expectations and agrees with your notes, being now suitable for publication in Materials Journal.

The answers were listed below:

Question # 1 - Certain state-of-art materials as anodes in LIB are missing, for example, Dual-transition-metal oxides. The authors need to add more references such as:

doi.org/10.1016/j.jallcom.2016.12.094

doi.org/10.1016/j.jallcom.2019.151691

Answer: We appreciate the suggestions. We add these references to Dual-transition-metal oxides suggested as general examples of the materials generally used in LIB, but without extending the discussion to not take the focus off the oxides produced by anodizing, which is the Material of interest in this manuscript and the Scopus of the Material´s special issue topic.

Question # 2 - Citations regarding AAO used in Li-S batteries are also missing, such as AAO as a template. Authors need to add those references and discussions.

Answer: This information and the references were added to the manuscript.
